# Global quantification of the dispersion effect with POLDER satellite data

Hengqi Wang ®[1,2], Yiran Peng ®[1] ✉, Antonio Di Noia ®[3], Huazhe Shang[2], Husi Letu[2], Bastiaan van Diedenhoven ®[4], Otto P. Hasekamp ®[4], Yangang Liu ®[5] & Johannes Quaas ®[6]

Increased aerosols can modify the shape of the cloud Particle Size Distribution (PSD), thereby influencing the radiative properties of clouds, known as the Dispersion Effect (DE). However, a global, observation-based quantification of its impact on Aerosol-Cloud Interactions (ACI) is lacking, leading to DE being typically ignored in satellite-based estimates of ACI forcing. Here we propose a physics-based method that combines polarimetric satellite data on cloud PSD to achieve global observational quantification of DE's impact on ACI in liquid-phase stratiform clouds. Globally, DE offsets ACI changes induced by droplet number concentration variation and liquid water path adjustment by 7% and −1.4%, respectively. Furthermore, a parameterization based on the global dataset of PSD shape parameters is developed to improve DE estimation in large-scale models. Both the quantification and parameterization enhance our understanding of DE and facilitate the inclusion of this non-negligible impact of DE on ACI in estimating aerosol climate forcing.

The increase in anthropogenic aerosols affects the radiative properties of liquid clouds by altering their microphysical (cloud droplet concentration, $N_d$, and effective radius, $R_e$) and macrophysical (liquid water path, LWP, and liquid cloud fraction, $f$) properties, phenomena generically known as aerosol–cloud interactions (ACI). Additional aerosols cause a monotonic increase in $N_d$ and, given a constant LWP, a decrease in $R_e$, leading to a net cooling effect on the earth-atmosphere system, referred to as the Twomey effect[1]. Subsequently, LWP and $f$ respond to changes in $N_d$, and $R_e$, known as rapid adjustments of ACI[2,3], which, together with the Twomey effect, contribute to estimating the effective radiative forcing due to ACI (ERF_aci). An example of such an adjustment is the cloud lifetime effect[4,5]. The latest report of the Intergovernmental Panel on Climate Change pointed out that ACI is one of the largest sources of uncertainty in current climate assessments[6].

In fact, besides $N_d$, the shape of the cloud particle size distribution (PSD) is also a key factor influencing the Twomey effect. Increased

aerosols change the cloud PSD shape, impacting cloud albedo and thus contributing to ERF_aci, referred to as the dispersion effect (DE). Liu and Daum[7] analyzed marine clouds sampled by aircraft and pointed out that the DE can offset the number effect (i.e., considering only the impact of $N_d$ changes on $R_e$ in the Twomey effect) by 10–80%. As research progressed, aircraft-based studies found such discrepant results that DE not only offsets ACI[8,9] but may also enhance it[10–13] or have no significant impact[14,15]. Considering the regional nature and significant uncertainty of the DE, this effect is typically ignored when estimating ERF_aci based on satellite observations[3,16].

By using parameterizations derived from the regional aircraft data in general circulation models (GCMs), the global impact of DE on ACI can be estimated. For instance, modelers applied different parameterizations[17–21] in GCMs and found DE can offset the number effect by −13–35%[9,17,22]; Xie et al.[23] incorporated three parameterizations[17,19,20] into a GCM and demonstrated that DE could offset ~7–14% of ACI globally. The high uncertainty in

[1]Department of Earth System Science, Ministry of Education Key Laboratory for Earth System Modeling, Institute for Global Change Studies, Tsinghua University, Beijing, China. [2]State Key Laboratory of Remote Sensing and Digital Earth, Aerospace Information Research Institute, Chinese Academy of Sciences, Beijing, China. [3]Institute of Environmental Physics, University of Bremen, Bremen, Germany. [4]SRON Space Research Organisation Netherlands, Leiden, Netherlands. [5]Environmental Science and Technologies Department, Brookhaven National Laboratory, Upton, NY, USA. [6]Leipzig Institute of Meteorology, Leipzig University, Leipzig, Germany. ✉e-mail: pyiran@mail.tsinghua.edu.cn

model evaluations may be related to the use of regionally dependent cloud PSD parameterizations. Additionally, GCM results can only provide the range of DE variations and generally lack observational validation on a global scale.

DE can be characterized as the sensitivity of the cloud PSD parameter (commonly represented by the ratio of effective radius to volume-mean radius, $\beta$) to the increasing aerosol number concentration ($N_a$)[24,25]. However, it is challenging to simultaneously obtain both $\beta$ and $N_a$ in clouds, so some cloud variables are commonly used as proxies for $N_a$. In the early stages, $N_d$ was widely used[9,17,20]. As research progressed, it was found that changes in the liquid water content per cloud droplet (i.e., $LWC/N_d$, hereafter LN) could better reflect DE, with a power–law relationship[19,26]:

$$\beta = a(LN)^b, \qquad (1)$$

where $a$ and $b$ are fitting parameters and DE is closely related to the parameter $b$[19]. Therefore, the key to quantifying DE is obtaining $\beta$ and LN, thereby determining parameter $b$.

The data of $\beta$ and LN used in previous studies were mainly obtained through regional aircraft observations[9,17,19,20,27], which makes DE quantification lack global representativeness. In satellite observations, the cloud PSD is typically described by $R_e$ and the effective variance ($V_e$), with $V_e$ being often fixed or discrete[28,29], which hinders the global analysis of DE. Recently, a global dataset with both $R_e$ and continuous $V_e$ available from the POLarization and Directionality of Earth's Reflectances (POLDER) instrument was established using artificial neural networks (NNs)[30] (hereafter, POLDER-NNs, see Methods), allowing us to provide a global estimate of the impact of DE on ACI.

Two steps are proposed for the global quantification (see Methods for more details):

Step 1: Using $R_e$ and $V_e$ retrieved by POLDER-NNs over global regions dominated by liquid-phase stratiform clouds, the cloud-top $\beta$ and LN can be determined as

$$\beta_P = \left[ (1 - V_e)(1 - 2V_e) \right]^{-1/3}, \ LN_P = A(1 - V_e)(1 - 2V_e)R_e^3. \qquad (2)$$

Here, the subscript $P$ indicates calculations based on the POLDER-NNs dataset, $A = 4\pi\rho_w/3$, where $\rho_w$ is the water density. By using $\beta_P$ and $LN_P$ for parameter fitting in Eq. (1), a cloud PSD parameterization based on global observations can be obtained.

Step 2: In adiabatic clouds, the cloud optical thickness ($\tau_c$) depends on cloud-top $\beta$ and $N_d$, as well as LWP ($\tau_c \propto \beta^{-1} LWP^{5/6} N_d^{1/3}$)[31]. By using the adiabatic liquid water lapse rate ($c_w$) to relate LWP to cloud-top LWC, $\tau_c$ can be fully expressed in terms of cloud-top variables, leading to $\tau_c \propto \beta^{-1} LWC^{5/3} N_d^{1/3}$. Given that $\beta$ follows Eq. (1), based on the calculation of $ERF_{aci}$[3], the impacts of DE on the number effect ($DO_{Nd}$, i.e., the impact of instantaneous DE) and the LWP adjustment effect ($DO_{LWP}$, i.e., the impact of adjusted DE) can be expressed as

$$DO_{Nd} = -3b \cdot 100\%, \ DO_{LWP} = \frac{3}{5}b \cdot 100\%, \qquad (3)$$

where DO stands for the dispersion offset (in %), and $b$ can be derived from step 1.

## Results and discussion
### Global distribution of DE's impact on ACI
First, we use POLDER-NNs data over global regions dominated by liquid-phase stratiform clouds to calculate the spatial distribution of the parameter $b$, and based on this, derive the global distributions of $DO_{Nd}$ and $DO_{LWP}$, as shown in Fig. 1.

The variables directly provided by the POLDER-NNs dataset are presented in Fig. 1a, b, including $R_e$ and $V_e$. The $\beta_P$ and $LN_P$ calculated from $R_e$ and $V_e$ are shown in Fig. 1c, d. Overall, the spatial distribution of $\beta_P$ is similar to that of $V_e$, while $LN_P$ is similar to $R_e$. Specifically, both $\beta_P$ and $LN_P$ exhibit distinct land–ocean distribution characteristics, which align with our general understanding of these variables (see Methods). By fitting a power–law relationship between $\beta_P$ and $LN_P$ within each grid point, the parameter $b$ can be determined (Fig. 1e). The dotted areas indicate that the fitting relationships are statistically significant with a 95% confidence level. Overall, the spatial distribution of the parameter $b$ exhibits two characteristics: (1) $b$ values across different grid points are predominantly negative; (2) there is significant spatial variability for the parameter $b$. Next, we conduct further analysis focusing on the two characteristics.

In detail, the proportion of negative $b$ values fitted using POLDER data exceeds 97% (with the dotted areas being 100%) (Supplementary Fig. 1). However, the parameter $b$ calculated through aircraft observations could be negative[19] or positive[26]. We think the discrepancy is likely due to the fact that the previous in situ measurements were primarily of local/regional scale with higher spatial resolution compared to the POLDER-NNs[30]. Aircraft in situ observations typically cover a range of kilometers, capturing fine-scale variations of $\beta$ to LN within clouds. Lu et al.[32] and Zhang et al.[33] demonstrated, through in situ observations and numerical simulations, that the cloud PSD parameter shows a positive correlation with LN for small cloud droplets. The relatively coarse spatial grid of the POLDER-NNs can only capture the dominant large-scale relationship, overshadowing the less frequent positive correlations and leading to our generally negative calculated $b$ values (Fig. 1e). However, since the grid scale used in GCMs is also relatively coarse and reflects the overall conditions within large-scale grids of hundreds of kilometers, these results are appropriately matched for model evaluations.

Additionally, there is significant spatial dependency in the distribution of $b$ values across different grid points (with a standard deviation of 0.015 and 0.013 within the dotted areas). Overall, more negative $b$ values are predominantly concentrated in regions heavily influenced by anthropogenic aerosols, indicating that the PSD shape response to aerosol changes is more sensitive in these regions. To explain the spatial distribution of $b$, we plotted the relative changes of $\beta_P$ and $LN_P$ (Supplementary Fig. 2). Analysis revealed that the parameter $b$ over land is determined by the combined variations of $\beta_P$ and $LN_P$, while over ocean, it is primarily determined by the variations in $\beta_P$ (see Methods). However, current GCMs do not consider spatial variations in the parameter $b$, which could introduce biases in simulating the cloud PSD and DE.

The spatial distribution of $b$ can be used to estimate the spatial distribution of $DO_{Nd}$ and $DO_{LWP}$ (Fig. 1f, g), indicating that the impacts of instantaneous and adjusted DE also exhibit spatial variability. Considering only regions with high-reliability $b$ values (dotted areas), DE globally exhibits a 9.31% offset on the number effect and a 1.86% enhancement on LWP adjustment effect.

### Overall assessment of DE's impact on ACI
Next, we examine all POLDER-NNs data collected throughout the year over global regions dominated by liquid-phase stratiform clouds. By fitting $\beta_P$ and $LN_P$ calculated from all the data, we find that they exhibit a clear power–law relationship (Pearson correlation coefficient $r = -0.53$, $p < 0.01$), with the fitting equation being $\beta_P = 0.68 LN_P^{-0.024}$ (Fig. 2a). According to Eq. (3) and $b = -0.024$, DE can offset 7.2% of the number effect but enhance the LWP adjustment effect by 1.44%.

Considering the impact of underlying surfaces on the cloud PSD, we further conduct regressions for land and ocean separately (Fig. 2b, c). The $b$ values are $-0.026$ for land and $-0.028$ for ocean. Correspondingly, the values of $DO_{Nd}$ and $DO_{LWP}$ are 7.8, $-1.56\%$ for land and 8.4, $-1.68\%$ for ocean. Specifically, the mean $LN_P$ for the land

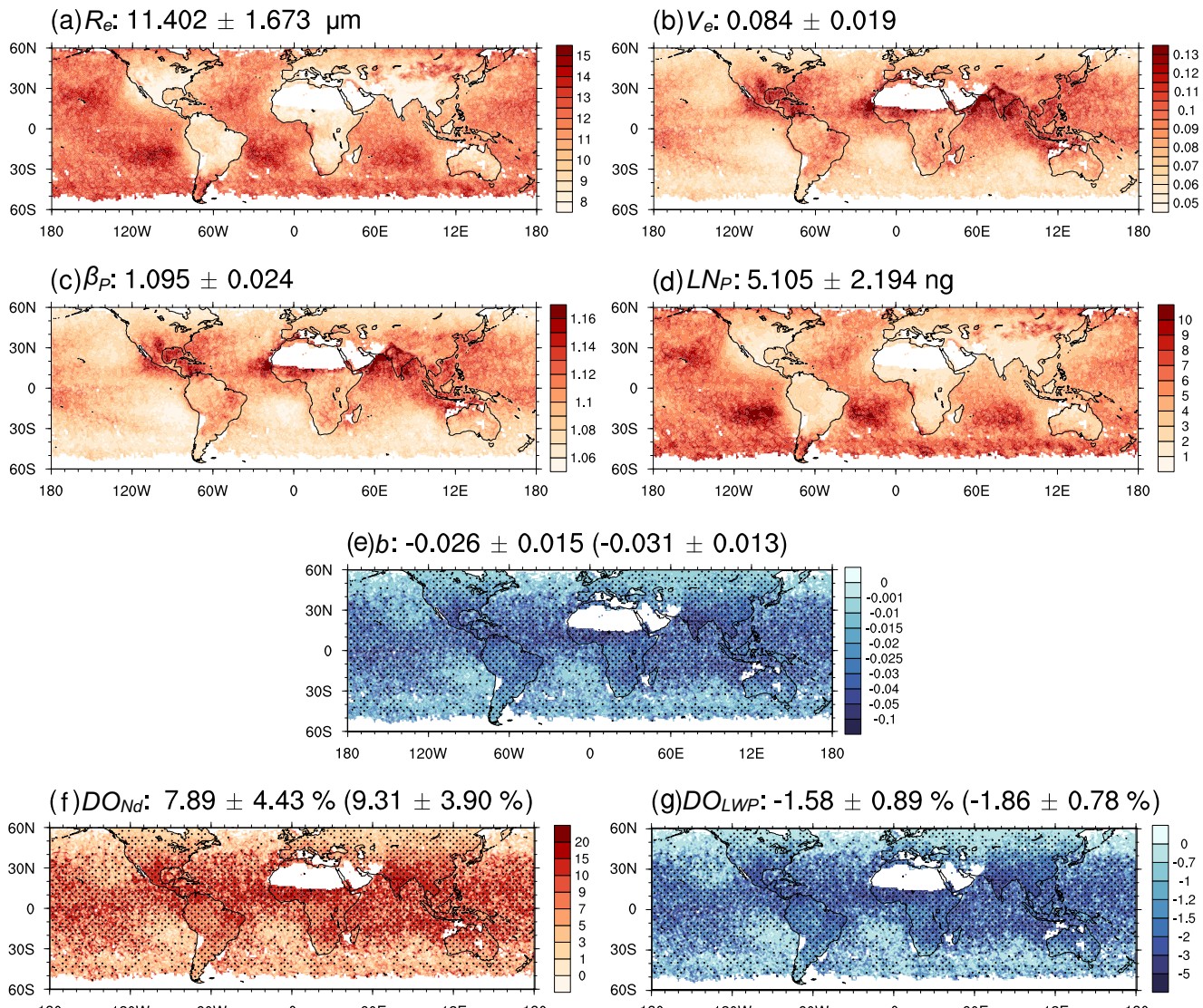

**Fig. 1 | Spatial distribution of annual mean values for variables related to satellite data (POLDER-NNs) used in this study.** The first row shows variables directly provided by the POLDER-NNs dataset, including **a** the effective radius ($R_e$) and **b** the effective variance ($V_e$). The second row shows variables calculated from $R_e$ and $V_e$, including **c** the particle size distribution parameter ($\beta_P$) and **d** the liquid water content per cloud droplet ($LN_P$). The third row presents **e** the parameter $b$ by fitting $\beta_P$ and $LN_P$ within each grid point (1° × 1°). The fourth row indicates variables derived from the parameter $b$, including the impacts of the dispersion effect on **f** the number effect ($DO_{Nd}$) and **g** the liquid water path adjustment effect ($DO_{LWP}$), where the dotted areas indicate the fitting relationships are statistically significant with a 95% confidence level. The global mean and standard deviation are shown in the title of each plot, with those for the dotted areas provided in parentheses.

(5.2 ng) is lower than that for the ocean (7.3 ng), but the mean $\beta_P$ for the land is slightly higher (1.098 vs. 1.095). Overall, the impact of underlying surfaces on the parameter $b$ is insignificant, changing the $b$ value by only about 0.002.

A linear regression in log-to-log space is applied to all the satellite data directly in the above analyses. However, a more widely used approach for calculating the sensitivity between two variables with huge amounts of satellite data is the pre-binned method[16,34–36]. Here, we also use the pre-binned method to fit the parameter $b$, as shown in Fig. 2d–f. The values of the parameter $b$ in global, land, and ocean regions are −0.020, −0.024, and −0.023, with |r| not less than 0.87, demonstrating a stronger power–law relationship. Comparing these with Fig. 2a–c, the absolute biases between the two methods for global, land, and ocean regions are 0.004, 0.002, and 0.005, respectively, suggesting that the calculation method of the fitting parameter influences the results, particularly in ocean regions. Additionally, the empirical power–law appears to fit the oceanic data better than the land data in the pre-binned method, which is more susceptible to

extreme values. We speculate that this is mainly related to retrieval algorithm challenges in accurately retrieving large cloud droplets. Compared to oceanic regions, retrieval uncertainty for large cloud droplets over land is greater[30]. This increased uncertainty may cause $\beta_P$ over land to become less sensitive to $LN_P$ with a value greater than 5 ng (see Methods), thereby weakening the fitted correlation coefficient (Fig. 2e).

### Quantitative analysis of uncertainty
Currently, the quantifiable sources of uncertainty include: (1) the inherent limitations of the POLDER-NNs[30]; (2) cloud heterogeneity[37]; (3) the retrieval method, wavelength, and grid scale[29]; and (4) the fitting method for parameter $b$[34]. Based on previous studies, we determined the uncertainty range of $b$ caused by different sources by considering a bias-corrected random Gaussian noise (Fig. 3a). Subsequently, a Monte Carlo method[38] was applied to evaluate the overall impact of the four uncertainty sources on the estimation of $b$. It was assumed that the $b$ values associated with each source follow a normal

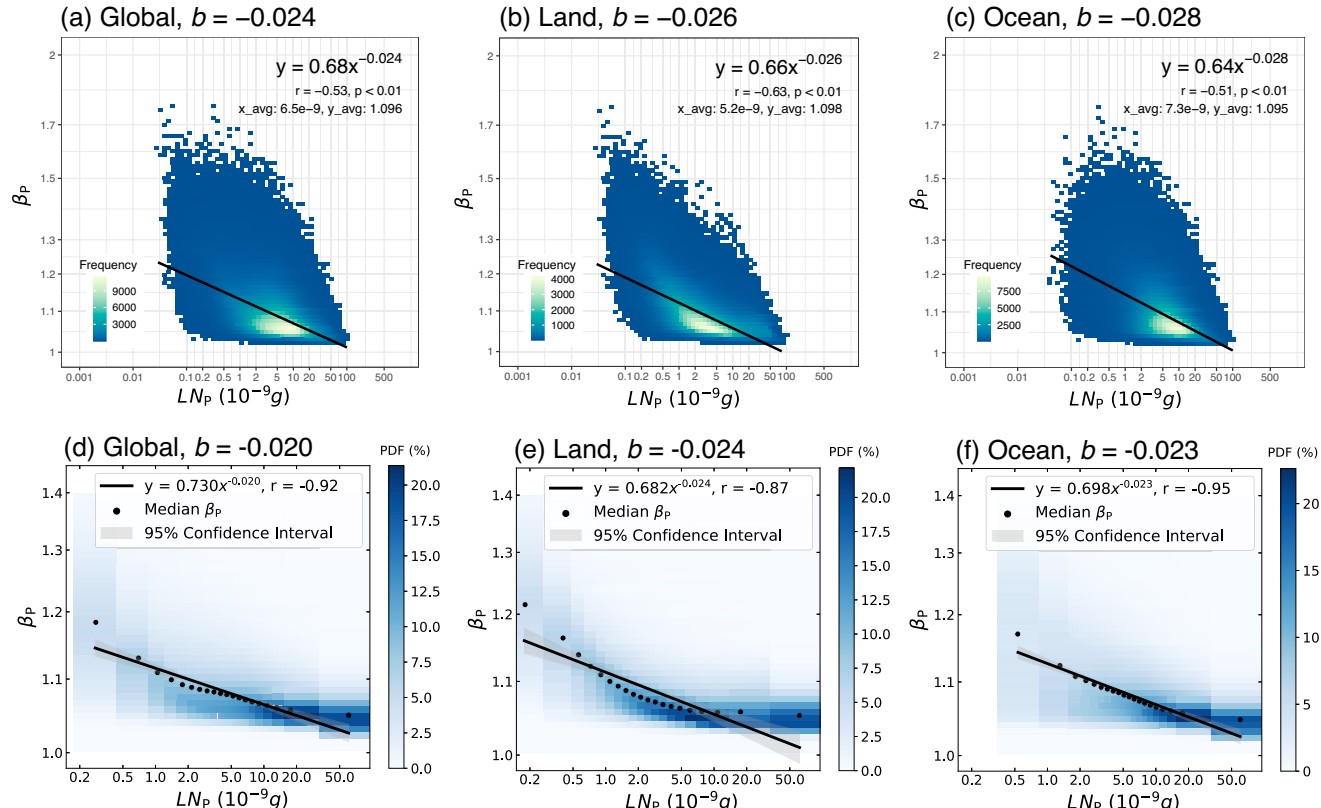

**Fig. 2 | Relationships between particle size distribution and liquid water content across regions.** The x-axis represents the particle size distribution parameter ($\beta_P$), while the y-axis represents the liquid water content per cloud droplet ($LN_P$). The first row shows scatter density plots, and the second row displays joint histograms, all plotted with logarithmic scales on both axes. Panels **a**, **d** correspond to the global region (60°S–60°N), **b**, **e** to land, and **c**, **f** to ocean. The solid black lines represent the fitted curves, and the fitting equations along with statistical parameters are labeled in the top right corner, where $y$ means $\beta_P$

and $x$ means $LN_P$. For the scatter density plots, the color represents the frequency of data points within each small interval. And $x\_avg$ and $y\_avg$ means the averages of $LN_P$ and $\beta_P$, respectively. For the joint histograms, the color represents the probability density within this range, wherein each column is normalized so that it sums to 1. The black dot represents the median of $\beta_P$ with an equal number of samples. The shaded area is the 95% confidence interval (according to a Student's $t$-test) to represent the fitting uncertainty.

distribution. A random sampling process was conducted 10 million times, and for each iteration, the mean $b$ value influenced by the different sources was calculated, resulting in a probability density distribution of $b$ (Fig. 3b). The mean of this distribution is taken as the best estimate of $b$, while the 5–95% confidence interval is used to represent the uncertainty range. A detailed description of the method is provided in the Methods section.

The results show that the best estimate of the $b$ (with 5–95% uncertainty in parentheses) is −0.024 (−0.026 to −0.022) globally, −0.025 (−0.028 to −0.022) over land, and −0.029 (−0.031 to −0.026) over ocean. The corresponding $DO_{Nd}$ values are 7.2% (6.6–7.8%), 7.5% (6.6–8.4%), and 8.7% (7.8–9.3%), and the $DO_{LWP}$ values are −1.44% (−1.56 to −1.32%), −1.50% (−1.68 to −1.32%), and −1.74% (−1.86 to −1.56%), respectively, as shown in the Table 1. This indicates that the dispersion effect caused by increased aerosols offsets the number effect by ~7% and enhances the LWP adjustment effect by about 1.4% globally, with a stronger impact on clouds over the ocean.

**Comparison with previous studies**

This section compares $b$ and DO obtained by POLDER-NNs with those by aircraft and GCMs (Table 1). In most studies, the $b$ values were derived from aircraft observations and DO values estimated in GCMs (an exception is the study by Liu and Daum[7], which derived DO from aircraft data and theoretical estimation).

The aircraft data used to fit $b$ in previous studies were conducted in various regions. For instance, Liu et al.[19] obtained a value of −0.14 by analyzing aircraft observations sampled in North America. Martins and

Silva Dias[26] fitted the relationship between $\beta$ and LN by studying clouds in the Amazon, obtaining a value of 0.072. In this study, whether fitting globally different regions as a whole (−0.026 to −0.022) or fitting data separately within each grid and then calculating the global mean (−0.031 ± 0.013), the $b$ values fall within the range of previously fitted $b$ values using aircraft observations (Table 1). Additionally, the $b$ values calculated in this study are predominantly negative across various global regions, and their absolute values are smaller compared to previous results.

Previous studies primarily focused on the impact of instantaneous DE (i.e., $DO_{Nd}$), as shown in Table 1. In the early stages, Liu and Daum[7] suggested that $DO_{Nd}$ could even reach 80%. However, as research progressed, the values for $DO_{Nd}$ consistently decreased. Recent research based on GCMs shows that the impact of instantaneous DE ranges from −13% (enhancement) to 35% (offset)[9,17,22]. Our results (6.6–7.8%) fall within the range of $DO_{Nd}$ calculated by previous studies. However, previous estimations could only provide a range of $DO_{Nd}$ calculated from different parameterizations, without validation with global observations. The $DO_{Nd}$ obtained in this study could be a validation reference for the assessment of parameterizations, contributing to further improvements and developments in GCMs.

Previous parameterizations derived from aircraft in situ observations are all at a local/regional scale, which partially meets the requirements for global simulations conducted by GCMs. This study utilizes satellite data to obtain a global-scale fitting parameterization ($\beta = 0.68LN^{-0.024}$ or $\beta = 0.73LN^{-0.020}$, see Fig. 2), which better meets the requirements for GCM applications. In the future, the

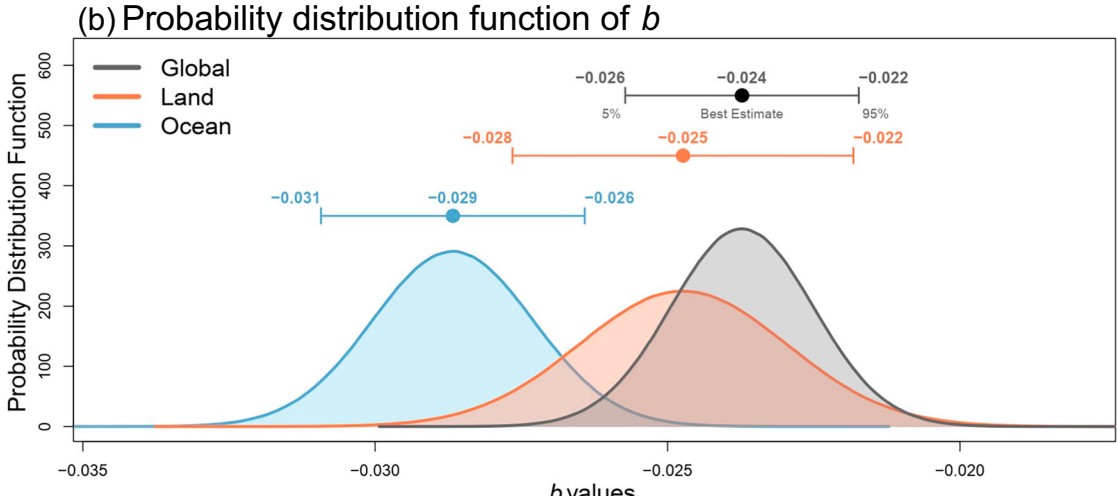

## (a) Flowchart of uncertainty quantification

## (b) Probability distribution function of $b$

**Fig. 3 | A framework of uncertainty quantification. a** The flowchart of uncertainty quantification, where $b_{s\_F}$ and $SE_{s\_F}$ represent the fitting parameter $b$ and its standard error (SE) considering different sources of uncertainty ($S = s1, s2, s3$, representing sources 1, 2, and 3) and using different fitting methods ($F = p, d$, representing pre-binned and direct fitting methods). $N(b_{s\_F}, SE_{s\_F}^2)$ represents a normal distribution with a mean of $b_{s\_F}$ and a standard deviation of $SE_{s\_F}$. **b** The probability distribution functions of the parameter $b$ in ocean, land, and global scales, where the point and errorbar represent the best estimate (i.e., the mean value) and its 5–95% confidence interval.

parameterization is expected to enhance the models' ability to simulate cloud PSD for liquid-phase stratiform clouds, thereby reducing the uncertainty in climate assessments.

To ensure the robustness and applicability of the conclusions, the following sections provide a detailed discussion of the assumptions, causal interpretation, limitations, and implications for future ACI studies.

### Assumption of adiabaticity of clouds
The derivation of DO is dependent on the assumption that the observed clouds are under moist adiabatic conditions, wherein clouds are not subject to the entrainment-mixing process. According to the

dependence of optical depth on the adiabatic liquid water lapse rate ($c_w$) in Eq. (15) as $c_w^{-1/6}$, variations in sub-adiabaticity due to entrainment-mixing are unlikely to significantly affect the albedo or the results of this study when modifying the overall $c_w$ in the cloud (Eq. (16)).

### Causal analysis of aerosol effects on LN and $\beta$
First, this causal relationship aligns with the physical understanding of cloud microphysical processes. The formation of the cloud PSD is primarily controlled by, until collision-coalescence sets in, the condensation growth process[19,21]. And the negative relationship basically reflects the fact that condensation leads to a narrow size distribution as

**Table 1 | Comparisons of results from this study, aircraft observations, and general circulation models (GCMs)**

| Data source | | $b$ | $DO_{Nd}$(%) | $DO_{LWP}$(%) |
|---|---|---|---|---|
| POLDER | Global | −0.024 (−0.026 to −0.022) | 7.2 (6.6–7.8) | −1.44 (−1.56 to −1.32) |
| | Land | −0.025 (−0.028 to −0.022) | 7.5 (6.6–8.4) | −1.50 (−1.68 to −1.32) |
| | Ocean | −0.029 (−0.031 to −0.026) | 8.7 (7.8–9.3) | −1.74 (−1.86 to −1.56) |
| Aircraft | Ref. 19 | −0.14 | / | / |
| | Ref. 26 | 0.072 | / | / |
| | Ref. 7 | / | 10–80 | / |
| GCMs | Ref. 9 | / | 33 | / |
| | Ref. 17 | / | 12–35 | / |
| | Ref. 23 | / | / | 7–14 |
| | Ref. 22 | / | −13–10 | / |

$b$ is the fitting parameter. $DO_{Nd}$ and $DO_{LWP}$ are the impacts of the dispersion effect on the number effect and the liquid water path adjustment effect, respectively. Values in parentheses represent the 5–95% confidence interval. The $b$ of ref. 26 is derived from the relationship between relative dispersion and the liquid water content per cloud droplet. The result of ref. 23. indicates the total impact of the dispersion effect on aerosol–cloud interactions.

droplets grow. The empirical relationship between LN and $\beta$ was demonstrated by Wood [39], which was later supported by the theoretical analysis of Liu et al. [40] and further validated by aircraft observations from several campaigns by Liu et al. [19]. A significant relationship obtained using POLDER-NNs to some extent validates this assumption (Figs. 1, 2). Therefore, we think the co-varying relationship between $\beta$ and LN has solid physics behind it, instead of being an observational artifact.

In addition, this causal relationship can also be confirmed through statistical analysis. To examine the relationship between aerosols, we employ a POLDER aerosol product [41–44] to calculate the aerosol index (AI) and plot joint histograms of AI versus LN and $\beta$ (See Methods). As shown in Fig. 4a, b, AI and LN exhibit a negative power–law relationship, whereas AI and $\beta$ show an overall positive power–law correlation, which is consistent with theoretical analysis. When aerosol increases and the LWC in the cloud remains relatively stable, the liquid water per cloud droplet (LN) decreases. At the same time, $\beta$ increases overall with AI, aligning with most previous studies that reported aerosol-induced broadening of the cloud PSD [7,9,19,20]. Given the negative power–law relationship between LN and $\beta$ (Fig. 2), it is hypothesized that LN plays a crucial mediating role in the impact of AI on $\beta$.

To investigate this, we conducted a causal mediation analysis [45] to examine the role of LN as a mediator in AI's impact on $\beta$, with the results shown in Fig. 4c (see Methods). The average causal mediation impact (0.0037) indicates the influence of AI on $\beta$ transmitted through the mediator LN, while the average direct impact (0.0041) represents the direct influence of AI on $\beta$. The total impact is 0.0078, and all results are statistically significant ($p < 0.001$). The results indicate that LN plays a significant mediating role in the impact of AI on $\beta$, further supporting the causality of the findings.

**Limitations of this study**
Some limitations of this study need to be pointed out. First, due to the limited data resolution and the predefined data filtering criteria, this study mainly focuses on liquid-phase stratiform clouds, which account for over 78% of the samples (see Methods and Supplementary Fig. 3). Accordingly, the cloud PSD analysis, quantification of the dispersion effect, and parameterization proposed in this work are primarily applicable to liquid-phase stratiform clouds. While some large-scale cumulus clouds are included, the relevance of our findings to typical

cumulus clouds remains uncertain and warrants further investigation using higher-resolution satellite observations [46–48] (e.g., from the multi-viewing multi-channel multi-polarization imaging [49]). Given that cumulus clouds generally play a less significant role in ACI [50], our focus aligns with the core objectives of current ACI research. Second, the analysis in this study is limited to 1 year of data (2006). Longer-term datasets are needed in the future for further validation and trend analysis. Finally, although the $\beta$ values calculated based on $V_e$ fall within the uncertainty range of previous studies, the $V_e$ in POLDER-NNs has only been compared with synthetic data and has not yet been validated against observational data. Despite these limitations, the quantitative method and analytical framework proposed in this study can still provide valuable insights for future DE research.

**Implications for future ACI estimation**
Additional aerosols can modify the shape of cloud PSD, thereby modifying the cloud albedo and forcing, a phenomenon known as the dispersion effect (DE). However, this effect is typically ignored when calculating the $ERF_{aci}$, largely due to a lack of global quantitative estimations of DE's impact on ACI. To address the gap, this study proposed a quantitative method based on physical mechanisms, utilizing physical equations and theoretical derivations. By using POLDER satellite observations, this study quantifies the global impact of DE on ACI over regions dominated by liquid-phase stratiform clouds. Based on a comprehensive analysis from multiple perspectives, it can be concluded that DE offsets the number effect by ~7% but enhances the LWP adjustment effect by around 1.4% on a global scale. Hence, the DE has a non-negligible impact on ACI, necessitating its consideration in future $ERF_{aci}$ calculation.

## Methods
### POLDER retrievals
The POLarization and Directionality of Earth's Reflectances (POLDER) instrument, here in its version mounted on the PARASOL (polarization and anisotropy of reflectances for atmospheric science coupled with observations from a Lidar) microsatellite [51], provides global cloud properties by multi-angle polarimetric observations [52–54]. Thanks to the polarimetric measurements, POLDER can retrieve two pieces of information on the cloud PSD at the cloud top, namely, besides the cloud effective radius ($R_e$), also the effective variance ($V_e$) [55,56].

Recently, Di Noia et al. [30] introduced a neural network algorithm (NNs) to utilize multi-angle and multi-wavelength polarimetric measurements from POLDER Level-1 data (5 km × 6 km) to retrieve high-resolution, numerically continuous $R_e$ and $V_e$. Comparisons with currently available POLDER datasets indicate that the algorithm possesses improved capabilities in retrieving $R_e$. Furthermore, the method can provide continuous values of $V_e$ from 0.03 to 0.35, which are often fixed or discrete in others [28,29]. To ensure the accuracy of the retrieval results, eliminate the influence of ice-/mixed-phase clouds, and facilitate subsequent data usage, ref. 30 performed strict data screening and re-gridding on the retrieved data, resulting in a 1° × 1° dataset containing only liquid cloud samples, referred to as L2-REGRID-REFF. The screening and processing of data include:

1. The cloudbow scattering angle range observed by POLDER is between 135° and 165°;
2. The total cloud cover is greater than 0.95;
3. The cloud phase index is less than 50 (excluding ice clouds and mixed-phase clouds);
4. The cloud-top pressure is higher than 600hPa (further excluding the influence of ice);
5. Data over the ocean is not affected by sun glint;
6. The data is from mid-to-low-latitude regions with ample sampling points (60° S ~ 60° N);
7. The data is re-gridded to the 1° × 1° grid with a daily (instantaneous) temporal resolution.

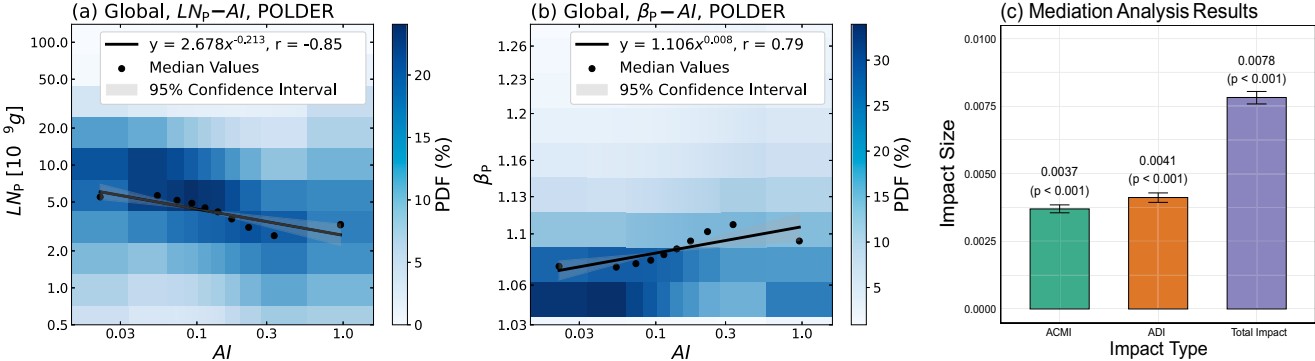

**Fig. 4 | Relationships between cloud parameters and aerosol, along with results of mediation analysis.** Joint histograms for **a** the liquid water content per cloud droplet ($LN_P$) and **b** the particle size distribution parameter ($\beta_P$) versus the aerosol index (AI) over the global region (60°S–60°N), with both axes on logarithmic scales.

**c** The result of mediation analysis, where ACMI and ADI represent the average causal mediation impact and average direct impact, respectively. The impact size means the change in $\ln(\beta)$ due to a one-unit increase in $\ln(AI)$. The bar represents the mean impact size, and the error bar indicates the 95% confidence interval.

Due to the large volume of Level-1 data (several terabytes) and the high computational burden of using the NNs algorithm, the L2-REGRID-REFF provided by Di Noia et al.[30] only includes results from 2006. To ensure the subsequent analysis results have statistical significance, we further excluded grid points with fewer than ten valid samples throughout the year. Finally, we obtained 1,127,994 valid samples, of which 400,776 are land data and 727,218 are ocean data. This dataset is referred to as POLDER-NNs and has been used in this study.

Cloud types in the POLDER-NNs dataset were classified based on the International Satellite Cloud Climatology Project (ISCCP) cloud classification scheme[57]. The results show that stratocumulus, altostratus, cumulus, altocumulus, nimbostratus, and stratus account for 51.2, 22.2, 15.8, 5.7, 2.8, and 2.4% of the samples, respectively, as illustrated in Supplementary Fig. 3. Notably, stratocumulus clouds, despite their partly cumulus-like appearance, are typically categorized as stratiform clouds due to their broad horizontal extent and limited vertical development[58]. In addition, due to the strict cloud cover criterion (cloud cover >0.95 within the 5 km × 6 km grid), the cumuliform clouds included here are limited to those with relatively large horizontal scales, while small, isolated cumulus clouds (e.g., significantly smaller than 5 km × 6 km) are excluded.

In summary, the results in this study are primarily applicable to liquid-phase stratiform clouds (stratocumulus, altostratus, nimbostratus, and stratus, collectively accounting for over 78% of the samples). While some large-scale cumulus clouds are included in the dataset, the applicability of our findings to typical cumulus clouds remains uncertain and warrants further investigation. Additionally, this study specifically focuses on the impact of the dispersion effect on ACI. Since cumulus generally play a less dominant role in ACI due to their short lifetimes, small spatial coverage, and weak coupling with large-scale radiative processes[50], our focus on liquid-phase stratiform clouds aligns well with the primary goals of current ACI research.

## $\beta$ and LN derived from POLDER retrievals

In GCMs, $R_e$ is generally parameterized through a PSD parameter $\beta$ that relates $R_e$ to the volume-mean radius ($R_v$)[59,60]:

$$R_e = \beta R_v = \beta \left( \frac{3LWC}{4\pi \rho_w N_d} \right)^{1/3}, \tag{4}$$

where LWC is the liquid water content (in kg m$^{-3}$), $\rho_w = 1000$ kg m$^{-3}$ is the water density, and $N_d$ is the number concentration of cloud droplets that are assumed spherical. $\beta$ can be well estimated by the relative dispersion ($\varepsilon$, defined as the ratio of the standard deviation to the

mean radius) by assuming a gamma distribution of the cloud PSD[60]:

$$\beta = \frac{(1+2\varepsilon^2)^{2/3}}{(1+\varepsilon^2)^{1/3}}, \tag{5}$$

$$\varepsilon = \frac{\sigma}{R_m}, \tag{6}$$

where $\sigma$ is the standard deviation, and $R_m$ is the mean radius. However, both $\sigma$ and $R_m$ cannot be obtained through prognosis in GCMs. In order to calculate $\beta$ in global models, different parameterizations were proposed.

In the early 21st century, parameterizations of the cloud PSD only considered the relationship between $\varepsilon$ or $\beta$ and $N_d$[9,17,20]. Later parameterizations gradually recognized the importance of LWC and began linking $\beta$ with LWC/$N_d$ (the liquid water mass per cloud droplet, hereafter LN), expressing their relationship in power–law form[19,26], as shown in Eq. (1). Additionally, two-moment cloud microphysics schemes in GCMs can prognosticate LWC and $N_d$ directly[61], making this form convenient for use in models.

According to the retrieval algorithm of POLDER, the PSD of liquid clouds ($\phi$) is characterized by a gamma distribution[52,62]:

$$\phi(r) = Cr_c^{(1-3V_e)/V_e} e^{(-r_c/(R_e V_e))}, \tag{7}$$

where $C$ is the intercept parameter, and $r_c$ is the cloud droplet radius. Correspondingly, $\sigma$ and $R_m$ can be expressed in terms of $R_e$ and $V_e$[62]:

$$\sigma_P = \sqrt{(1-2V_e)V_e} R_e, \tag{8}$$

$$R_{mP} = (1-2V_e)R_e, \tag{9}$$

where the subscript $P$ indicates calculation using POLDER data.

Furthermore, $\varepsilon$ and $\beta$ can be derived from POLDER retrievals as

$$\varepsilon_P = \frac{\sigma_P}{R_{mP}} = \sqrt{\frac{V_e}{1-2V_e}}, \tag{10}$$

$$\beta_P = \frac{(1+2\varepsilon_P^2)^{2/3}}{(1+\varepsilon_P^2)^{1/3}} = \left[ (1-V_e)(1-2V_e) \right]^{-1/3}, \tag{11}$$

It should be noted that the mode radius of the gamma function is $(1-3V_e)R_e$[52], which means that the $V_e$ should be less than 1/3 to

ensure the existence of peaks in the gamma function (i.e., the mode radius is greater than 0). In the data provided by Di Noia et al. [30], the range of $V_e$ is given as 0.03 to 0.35. When conducting subsequent studies, the parts greater than 1/3 were excluded first.

According to Eqs. (4) and (11), we can obtain cloud-top LN using $R_e$ and $V_e$ provided by POLDER (denoted as $LN_P$):

$$LN_P = \frac{4\pi\rho_w}{3}(1 - V_e)(1 - 2V_e)R_e^3. \tag{12}$$

Exponentially fitting the $\beta_P$ and $LN_P$ yields parameter $b$ (Eq. (1)), which can be used to quantitatively assess the global impact of DE on ACI, as described below.

## Impacts of DE on ACI

The effective radiative forcing of aerosol–cloud interactions ($ERF_{aci}$) can be represented as the forcing sum of the Twomey effect (instantaneous effect) and the associated rapid adjustments[3,16]:

$$ERF_{aci} = F_{Nd} + F_{LWP} + F_f, \tag{13}$$

where $F_{Nd}$, $F_{LWP}$, and $F_f$ are the radiative forcing of the Twomey effect, and the radiative adjustments of LWP, and $f$, respectively.

Considering the dependence of cloud albedo ($\alpha_c$) on cloud optical depth ($\tau_c$)[63] and the relationship between $\tau_c$ with LWP and $N_d$[31], there are relationships in adiabatic clouds[3,64]:

$$\frac{d\ln\alpha_c}{d\ln\tau_c} = 1 - \alpha_c, \tag{14}$$

$$\tau_c = \frac{6}{5}\pi Q_{ext} B^{\frac{2}{3}}\beta^{-1}LWP^{\frac{5}{6}}N_d^{\frac{1}{3}}, \tag{15}$$

where $B = [(4/3)\pi\rho_w]^{-1}(2c_w)^{-1/4}$, $c_w$ is the adiabatic liquid water lapse rate ($c_w = dLWC/dz$, $z$ is the height above the cloud base) and is considered to be constant through the cloud[64], $Q_{ext}$ is the Mie efficiency factor and is usually set to 2[31,65], $\beta$ is often set as a constant in $ERF_{aci}$ calculation[3,16], and $N_d$ is assumed vertically uniform in adiabatic clouds. Thus, Eq. (15) can be rewritten as

$$\tau_c \propto LWP^{\frac{5}{6}}N_d^{\frac{1}{3}}. \tag{16}$$

Correspondingly, $F_{Nd}$ and $F_{LWP}$ can be represented as[3]

$$F_{Nd} = \frac{1}{3}\alpha_c(1 - \alpha_c) \cdot c_{Nd} \cdot \Delta\ln N_{d,ant}, \tag{17}$$

$$F_{LWP} = \frac{5}{6}\alpha_c(1 - \alpha_c) \cdot c_{LWP} \cdot \Delta\ln LWP_{ant}, \tag{18}$$

where $\Delta\ln N_{d,ant}$ and $\Delta\ln LWP_{ant}$ are the anthropogenic perturbations of $N_d$ and LWP. And $c_{Nd}$ and $c_{LWP}$ are the effective cloud fractions for $N_d$ and LWP, respectively. Its "effectiveness" stems not solely from the partial coverage offered by liquid clouds but also from considering the spatial correlations among other pertinent factors in deriving $F_{Nd}$ and $F_{LWP}$[3]. It should be noted that $F_{Nd}$ here only considers the impact of $N_d$ changes on $R_e$ and consequently $\alpha_c$, without accounting for $\beta$. Therefore, it can be regarded as part of the Twomey effect (referred to as the number effect).

In adiabatic liquid clouds, the vertical profile of LWC is termed the adiabatic condensation profile with a constant $c_w$:

$$LWC(z) = \int_0^z c_w dz = c_w z, \tag{19}$$

and for LWP:

$$LWP = \int_0^H c_w z dz = \frac{1}{2}c_w H^2, \tag{20}$$

where $H$ is the cloud depth and $c_w$ is approximately $2\,mg\,m^{-3}\,m^{-1}$. Considering that satellite observations mainly capture information at the cloud top, here we relate Eq. (16) to the cloud-top LWC and $N_d$, denoted as $LWC_{top}$ and $N_{d\_top}$, respectively. Since $N_d$ remains constant in the adiabatic cloud, $N_{d\_top}$ equals $N_d$. Utilizing Eq. (19), we can derive $LWC_{top}$ as:

$$LWC_{top} = c_w H. \tag{21}$$

Combining Eqs. (20) and (21), we obtain:

$$LWP = \frac{LWC_{top}^2}{2c_w}. \tag{22}$$

Substituting Eq. (22) and $N_{d\_top}$ into Eq. (16), we get:

$$\tau_c \propto LWC_{top}^{\frac{5}{3}}N_{d\,top}^{\frac{1}{3}}, \tag{23}$$

Although Eq. (23) incorporates $\beta$ in the definition of $\tau_c$ (Eq. (15)), $\beta$ is treated as a fixed parameter during practical implementation. As a result, variations in $\beta$ cannot be accounted for when evaluating $\tau_c$ under anthropogenic aerosol perturbations (Eq. (16)), thereby neglecting the influence of the dispersion effect in the estimation of $ERF_{aci}$ (Eqs. (17, 18)). When the dispersion effect is considered (i.e., when $\beta$ is treated as a variable rather than a constant), $\tau_c$ can be represented as[64]:

$$\tau_c \propto \beta^{-1}LWP^{\frac{5}{6}}N_d^{\frac{1}{3}}. \tag{24}$$

Given $\beta = a(LWC_{top}/N_{d\_top})^b$ (Eq. (1)), and utilizing Eq. (23), we now can rewrite $\tau_c$ in terms of $LWC_{top}$, $N_{d\_top}$, and the $b$ parameter as

$$\tau_c \propto LWC_{top}^{\frac{5}{3} - b}N_{d\,top}^{\frac{1}{3} + b}. \tag{25}$$

Correspondingly, $F_{Nd}$ and $F_{LWP}$, when considering the dispersion effect (i.e., $F_{N_d}^{disp}$ and $F_{LWP}^{disp}$), can be written as

$$F_{N_d}^{disp} = \left(\frac{1}{3} + b\right) \cdot \alpha_c(1 - \alpha_c) \cdot c_{Nd} \cdot \Delta\ln N_{d,ant} = F_{Nd} + 3bF_{Nd}, \tag{26}$$

$$F_{LWP}^{disp} = \left(\frac{5}{6} - \frac{b}{2}\right) \cdot \alpha_c(1 - \alpha_c) \cdot c_{LWP} \cdot \Delta\ln LWP_{ant} = F_{LWP} - \frac{3}{5}bF_{LWP}, \tag{27}$$

where $3bF_{Nd}$ and $-3/5 \cdot bF_{LWP}$ represent the radiative forcing caused by the instantaneous DE ($F_{DE\_Nd}$) and the adjusted DE ($F_{DE\_LWP}$). Accordingly, the total radiative forcing caused by the dispersion effect ($F_{DE}$) can be expressed as:

$$F_{DE} = F_{DE\_Nd} + F_{DE\_LWP} = 3bF_{Nd} - \frac{3}{5}bF_{LWP}. \tag{28}$$

To quantify the impact of the dispersion effect on ACI, we define the dispersion offset (DO):

$$DO_{Nd} = -\frac{F_{DE\_Nd}}{F_{Nd}} \cdot 100\% = -3b \cdot 100\%, \tag{29}$$

$$DO_{LWP} = -\frac{F_{DE\_LWP}}{F_{LWP}} \cdot 100\% = \frac{3}{5}b \cdot 100\%, \tag{30}$$

where $DO_{Nd}$ and $DO_{LWP}$ indicate the impact of DE on the number effect (i.e., the impact of instantaneous DE) and LWP adjustment effect (the impact of adjusted DE), respectively. Positive values indicate off-setting, while negative values indicate enhancement. The parameter $b$ can be obtained by fitting the $\beta_P$ and $LN_P$ from POLDER-NNs, as described in Eqs. (11) and (12).

## Explanation of the spatial distributions of $\beta_P$ and $LN_P$
In the main text, we pointed out that $\beta_P$ and $LN_P$ exhibit distinct land−ocean distribution characteristics. Here, we further discuss these characteristics.

The values of $\beta_P$ over land are generally higher than those over ocean, reflecting the broadening effect of aerosols on the cloud PSD, which is consistent with previous analyses based on aircraft observations[8,9]. However, even within the same oceanic or terrestrial region, $\beta_P$ exhibits regional variations. For instance, over oceans, coastal areas, and regions affected by aerosols (such as the North Pacific) have higher $\beta_P$ values compared to more open ocean areas (like the Southern Ocean). Similarly, over land, $\beta_P$ also shows regional differences. For example, $\beta_P$ values are higher in East Asia and India compared to the relatively cleaner Europe. Notably, $\beta_P$ is higher over India than over the more polluted regions of China, which we think could be related to the abundant moisture conditions in India, though the specific mechanisms need further investigation. The $LN_P$, which physically represents the amount of water vapor each cloud droplet can obtain, also shows land−ocean distribution characteristics. As shown in Fig. 1d, $LN_P$ values are generally lower over land compared to ocean, which is associated with relatively less moisture and higher aerosol concentrations over land.

## Explanation of the spatial distribution of the parameter $b$
The distribution of the parameter $b$ varies significantly across different grid points (standard deviation of 0.015, with 0.013 in the dotted areas), exhibiting distinct spatial distribution characteristics. Overall, regions with $|b|$ are primarily located in areas heavily influenced by anthropogenic aerosols, such as ocean regions near continental coastlines, the northern Indian Ocean, the North Atlantic, the North Pacific, and low-latitude regions of the South Pacific, as well as land areas including East Asia, South Asia, the Indian subcontinent, central Africa, and southern North America. This indicates that the cloud PSD in these regions is more sensitive to changes in aerosols.

To explain the spatial distribution of the parameter $b$, we plotted the relative changes in $LN_P$ and the corresponding $\beta_P$ (denoted as $\Delta\ln(LN_P)$ and $\Delta\ln(\beta_P)$, respectively) as shown in Supplementary Fig. 2a, b. The $\Delta\ln(LN_P)$ represents the difference between the mean $\ln(LN_P)$ of the highest 10% and the lowest 10% within a grid point, while $\Delta\ln(\beta_P)$ represents the corresponding difference in $\ln(\beta_P)$. With a constant LWC, an increase in LN indicates a decrease in cloud droplet/aerosol number concentration. Mathematically, using the principle of invariance of differential forms, we have $d\ln(LN_P) = d(LN_P)/LN_P$, meaning $\Delta\ln(LN_P)$ represents the relative change in average cloud droplet water content due to a decrease in aerosols. Similarly, $\Delta\ln(\beta_P)$ indicates the relative change in cloud PSD due to a decrease in aerosols.

By slightly transforming the power−law relationship between $\beta$ and LN (Eq. (1)), we get $b = d\ln(LN)/d\ln(\beta)$. We attempt to approximate $b$ using $\Delta\ln(LN_P)/\Delta\ln(\beta_P)$ (Supplementary Fig. 2c), thereby explaining the spatial variation of parameter $b$ through the spatial distributions of $\Delta\ln(LN_P)$ and $\Delta\ln(\beta_P)$. Comparing Supplementary Fig. 2c and Fig. 1e, the ratio of $\Delta\ln(LN_P)$ and $\Delta\ln(\beta_P)$ shows a similar global mean and spatial distribution to the parameter $b$, indicating that our analytical method is reasonable.

In general, the variation of $\Delta\ln(LN_P)$ exhibits a clear land−ocean distribution pattern. Compared to land regions, oceanic regions have generally lower values with less pronounced regional distribution features (Supplementary Fig. 2a). However, the variation of $\Delta\ln(\beta_P)$ in oceanic regions shows distinct regional distribution characteristics, with larger absolute values mainly occurring in the northern Indian Ocean, the North Atlantic, the North Pacific, and low-latitude regions of the South Pacific, ultimately leading to larger absolute values of $b$ in these areas. This indicates that the spatial distribution of the fitting parameter $b$ in ocean regions is primarily determined by the relative variation of $\beta_P$, while the relative variation of $LN_P$ across different regions is not significant.

In contrast to oceanic regions, the spatial distribution of the fitting parameter $b$ in land areas is determined by the relative changes in both $\beta_P$ and $LN_P$. For instance, in the Indian subcontinent, a relatively small relative change in $LN_P$ (Supplementary Fig. 2a) leads to a larger relative change in $\beta_P$ (Supplementary Fig. 2b), resulting in a more negative $b$ in this region. In Europe, although the relative change in $LN_P$ is similar to that in the Indian subcontinent, the relative change in $\beta_P$ is relatively weaker, leading to a smaller absolute value of the parameter $b$. Similar results are observed in other land areas. Overall, in land regions, the fitting parameter $b$ is determined by the combined relative changes in $\beta_P$ and $LN_P$.

It is important to note that due to the limited variables provided by the POLDER-NNs dataset, our analysis remains relatively coarse, and further analysis of the physical mechanisms is necessary in the future.

## Factors influencing the $\beta$ − LN relationship over land and ocean
In this study, we consider LN to be the dominant factor influencing $\beta$, and their relationship can be represented by a power−law relationship. However, when LN is either small or large, the relationship between $\beta$ and LN appears to deviate from this power−law fitted using the pre-binned method, particularly over land (Fig. 2e). In other words, when LN reaches extreme values, $\beta$ may be significantly affected by other factors. We speculate that this is related to the following three factors:

1. Bias of the retrieval algorithm: Compared to the ocean, the retrieval uncertainty, especially for large cloud droplets over land is greater[30]. Further analysis suggested that this may be linked to the influence of aerosols above clouds in these regions on the POLDER-NNs method[30]. This effect may cause $\beta$ over land to become insensitive to LN when greater than 5 ng (Fig. 2e), thereby reducing the fitted correlation coefficient.

2. Condensation (evaporation) occurring simultaneously with new activation (deactivation) for small droplets: Lu et al.[32] and Zhang et al.[33] demonstrated through in situ observations and numerical simulations that $\varepsilon$, which is proportional to $\beta$, shows a positive correlation with LN for small cloud droplets. This may deviate the $\beta$ − LN relationship away from a negative correlation. Compared to oceanic clouds, cloud droplets over land tend to be smaller overall, which may contribute to the different fitting performance in land and ocean.

3. Collision-coalescence process for large droplets: When LN is large enough, the collection process (precipitation initiation) may occur[66], thereby disrupting the original characteristics of $\beta$. Considering that the retrieval of large cloud droplets over land is inherently less accurate, this effect may be further exacerbated.

The above hypotheses regarding the land-ocean differences require further in-depth investigation for validation.

## Process of uncertainty quantification
The first source of uncertainty (Src1) arises from the inherent uncertainty in the POLDER-NNs method. The POLDER-NNs dataset employed in this study is generated via a neural network algorithm. Discrepancies between this dataset and the exact solution derived from the radiative transfer model introduce uncertainty in the retrievals of $R_e$ and $V_e$. Based on Table 3 from Di Noia et al.[30], Src1 is found to cause a bias (Bias) and root mean square error (RMSE) in $R_e/V_e$ of

0.08/−0.01 and 0.92/0.03, respectively, as shown in Supplementary Table 1. Based on this, we introduce a bias-corrected random Gaussian noise for each $R_e/V_e$ value in POLDER-NNs, denoted as $N(-Bias, RMSE)$, where $N$ represents a normal distribution with a mean of −Bias and a standard deviation of $RMSE^{1/2}$. Using the corrected values, we calculate $\beta$ and $LN$, then fit the data to obtain the parameter $b$ and its corresponding standard error (SE) while accounting for the impact of Src1. To ensure the robustness of the results and avoid potential biases from a single random sampling, we repeat this process 10,000 times (Fig. 3a). The final estimates of $b$ and SE, considering Src1, are obtained by averaging all sampled results and are denoted as $b_{s1d}$ and $SE_{s1\_d}$, where s1 refers to Src1 and $d$ indicates the use of the direct fitting method (Supplementary Table 1).

The second source of uncertainty (Src2) stems from the impact of cloud heterogeneity. The POLDER-NNs data utilized in this study are characterized by a relatively coarse resolution, which may introduce errors by assuming homogeneity within the retrieval area (-6 km resolution). Therefore, it is essential to account for the effects of cloud heterogeneity on the retrievals of $R_e$ and $V_e$. Shang et al.[37] evaluated this impact by modeling a cloud field comprising several equal-area subregions with constant cloud optical thickness but varying $R_e$ and $V_e$ values. Based on the differences between the actual and retrieved $R_e/V_e$ (as shown in Table 2 in Shang et al.[37]), Src2 is found to cause the Bias and RMSE in $R_e/V_e$ of −0.71/0.02 and 0.88/0.04, respectively (Supplementary Table 1). Following the uncertainty quantification framework from Src1, the uncertainty in $b$ caused by Src2 is then determined, and the $b_{s2\_d}$ and $SE_{s2\_d}$ are provided, as shown in Supplementary Table 1.

The third source of uncertainty (Src3) arises from the use of the POLDER retrieval method, wavelength selection, and grid-scale processing. The POLDER-NNs data are derived by applying machine learning to the multi-angle, multi-wavelength polarimetric measurements with a grid scale of 1° × 1°. Shang et al.[29] introduced an enhanced primary cloudbow retrieval (PCR) algorithm to estimate $R_e$ and $V_e$ from POLDER, creating a global retrieval dataset for four months (February, May, August, and November 2008) (denoted as POLDER-PCR). Unlike POLDER-NNs, POLDER-PCR employs traditional retrieval methods, clearly differentiating between various retrieval wavelengths (670/865 nm), with a grid resolution of 0.7° × 0.7°. Data from POLDER-PCR for low- and mid-latitude regions (60°S–60°N) were selected for fitting parameter $b$. Based on the fitting outcomes across two wavelengths, the uncertainty in $b$ due to Src3 is determined, and the corresponding $b$ and $SE$ are provided (denoted as $b_{s3\_d}$ and $SE_{s3\_d}$), as shown in Supplementary Table 1. Although POLDER-NNs and POLDER-PCR utilize different wavelengths and grid scale configurations, as well as distinct retrieval methodologies, the derived $b$ values are relatively consistent. Compared to POLDER-NNs, the $b$ values obtained from POLDER-PCR exhibit larger SE, likely due to the smaller sample size of the POLDER-PCR dataset (68,555 valid samples) and discontinuous $V_e$ values (intervals of 0.02). However, this convergence in $b$ values obtained from two independent retrievals provides a basis for evaluating the potential influence of Src3.

The fourth source of uncertainty (Src4) arises from the fitting method used for parameter $b$. As discussed before, commonly used fitting methods for large satellite datasets include direct and pre-binned fittings, though which of the two is the superior method remains unclear. Therefore, both fitting methods were employed to derive $b$ and $SE$ using Src1–Src3, as shown in Supplementary Table 1.

Finally, the overall impact of four sources on the estimation of $b$ is quantified using a Monte Carlo method, similar to that of Boucher and Haywood[38] and Bellouin et al.[3]. It is assumed that the $b$ values induced by different sources, as shown in Supplementary Table 1, follow a normal function (i.e., $b \sim N(b_{SF}, SE_{SF}^2)$, where $S$ represents different sources, and $F$ indicates the fitting method). A random sampling process is then performed 10 million times, and for each random result, the mean value of $b$ affected by different sources is calculated, resulting in a probability density distribution of $b$. The mean is then computed as the best estimate of $b$ (−0.024), and the 5−95% confidence intervals are determined as the uncertainty range for $b$ (−0.026 to −0.022), as shown in Fig. 3b and Supplementary Table 1.

Similar uncertainty quantification has also been applied to clouds over ocean and land, with the relevant parameters listed in Supplementary Table 2 and the probability density distributions shown in Fig. 3b.

**Causal mediation analysis**

To examine the relationship between aerosols, LN, and $\beta$, as well as the mediating effect of LN on the impact of aerosols on $\beta$, we use the POLDER Level 3 aerosol products, generated using a generalized retrieval of atmosphere and surface properties "components" approach, gridded at a 1° × 1° resolution (POLDER-3/GRASP, version 1.1)[41–44]. This product is officially recommended for studies involving both the Ångström exponent (AE) and aerosol optical depth (AOD) and demonstrates a high consistency with the aerosol robotic network (AERONET) on a global scale[44,67].

To better characterize the properties of aerosols that can be activated as cloud droplets, this study uses AE and AOD at 565 nm from the POLDER-3/GRASP to calculate the aerosol index (AI, $AI = AE \times AOD$)[34]. Before analysis, AI were matched with POLDER-NNs on a daily basis within a 1° × 1° grid to ensure full spatiotemporal alignment between the two datasets. The results of the matched data analysis are shown in Fig. 4, where the power−law fitting of LN and $\beta$ with AI is performed using the pre-binned method, and the mediation effect analysis is conducted using the R package for a causal mediation analysis[45].

## Data availability

The POLDER-NNs dataset used in this study can be downloaded publicly from ftp://ftp.sron.nl/open-access-data/antonion/10.5194-amt-2018-345 (last access: 4 January 2024). POLDER-3/GRASP dataset is from "CNES/GRASP/LOA/Cloudflight/ICARE" (last access: 10 March 2025).

## Code availability

The code used to replicate the figures in this study is available at https://doi.org/10.6084/m9.figshare.28648598[68].

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

## Acknowledgements
The authors thank the POLDER team for their contributions. This work is supported by the National Natural Science Foundation of China (42175096, Y.P. and 42025504, H.L.), the Hebei province Key Research and Development Project (20375402D, Y.P.), the National Key Research and Development Program of China (2023YFB3905900, H.S.), the China Scholarship Council (H.W.), and the National Key Scientific and Technological Infrastructure project "Earth System Numerical Simulation Facility" (EarthLab) (Y.P.). Y.L. is supported by the US Department of Energy Office of Science Biological and Environmental Research as part of the Atmospheric System Research (ASR) Program. J.Q., B.D., and O.P.H. acknowledge funding by the European Union Horizon Europe project "CleanCloud" (101137639).

## Author contributions
H.W., Y.P., and J.Q. designed this study. H.W. performed analyses and prepared the manuscript. Y.P. and J.Q. led this work and revised the paper. A.D.N. and H.S. provided the POLDER data and advised on the use of the data. H.L., B.D., O.P.H., and Y.L. gave instructive suggestions to improve the paper. All authors contributed to the final paper.

## Competing interests
The authors declare no competing interests.
