## [Transparent Peer Review file · Nature Communications]

Global Quantification of the Dispersion Effect with POLDER Satellite Data

Corresponding Author: Professor Yiran Peng

Version 0:

Reviewer comments:

Reviewer #1

(Remarks to the Author)

The noteworthy result in this paper is a global, observation-based quantitative estimation of the impact of the dispersion effect (DE) on Aerosol-Cloud Interactions (ACI), specifically the impact of the DE on changes to the planetary albedo that are caused by droplet number concentration changes and liquid water path adjustments. In particular, the way in which the authors formulate the dispersion effect as a fractional offset to the radiative forcing caused by ACI allows it to be applied to a wide range of existing results on ACI. The new dataset from POLDER, that provides both effective radius and effective variance, offers the first set of global observations that can be used to constrain the DE and the paper presents the benefits of this new global dataset and the potential limitations caused by the spatial scale at which it is available clearly. There are no obvious flaws in the analysis, interpretation and conclusions and specific suggestions and corrections to errors, or the identification of conflicting statements, are given below. The methods used are adequately presented in the Methods section, but the clarity of presentation in the body of the paper leaves something to be desired and suggestions for improvement are given below.

One of the most interesting figures to this reviewer is Fig. 2, and in particular the difference between Fig 2e) and 2f). The empirical power law appears to be a much better fit to the oceanic data than to the data over land (15% more variance explained). It would be of interest to hear whether the authors have a good explanation for this.

Detailed Comments

The parameter beta is introduced at line 55, but is not defined. Given that this parameter is used in many different ways it should at least be described as the ratio of effective radius to volume mean radius, even if an equation is not given.

The empirical equation linking beta with the ratio of Liquid Water Content (LWC) to droplet number concentration (Nd) is given in the text at line 61, but is not numbered, and cannot therefore be referenced. The order of the three steps proposed to quantify the impact of DE on ACI (line 71) makes no sense. After beta and (LWC/Nd) are estimated in step 1 the next step is surely to estimate a and b using equation (M4). Equation (M4) should therefore be listed here in the Main text so that it can be easily referenced as it is a key part of the analysis. Step 3 should therefore become Step 2 and can be properly described, rather than have the current vague one line description. In the current Step 2 (which should become Step 3) an equation is given providing the dependence of cloud optical depth on cloud top variables and then referring to Method 3. I strongly suggest also noting the form of the proportionality to LWP and reference eq.(17) of Reference 53. Reference 53 provides one of the earliest and most detailed rationales for the use of an adiabatically stratified framework in analyses of this type and making the link to this reference early in the presentation would be helpful to the reader. It also lists the constants of proportionality including the dependence on the adiabatic liquid water lapse rate.

In Eq.(3) it would be helpful to the reader if rather than using $DO_LWP=0.6*100\%$ it was written as $DO_LWP=3/5*100\%$, so that the interested reader can easily make the link to the power laws given for the dependence of cloud optical depth on cloud top variables.

The phrase “different grids” is used in several places (e.g. lines 101, 117). If you mean “different grid points”, or “different spatial locations” please say so.

On lines 139 and 140 you state that “for the ocean (5.2 ng) is lower than that for land (7.3 ng)”. The exact opposite is the case

shown in Figs. 2b) and 2c). I believe the statement in the text at lines 139/140 is wrong. Please correct text, or figure appropriately.

At lines 147/148 you state that “the absolute bias between the two methods for global, land, and ocean regions are 0.04, 0.02, and 0.05, respectively”. Examining Fig. 2 suggests these values are all too large by an order of magnitude. Please correct.

In the discussion of adiabatic conditions, it is probably worth referring back to the dependence of optical depth on adiabatic liquid water lapse rate given in eq.(17) of Ref 53 which is $Cw^{-1/6}$. This suggests that variations in how sub-adiabatic a profile is, as a result of entrainment mixing, is not going to have a substantial impact on the albedo, or the results presented in this paper.

At line 202 the statement is made that “This empirical finding is corroborated by the theoretical analysis conducted by Liu et al.39”. That Liu et al. paper makes the link between dispersion and number concentration, but to say that theoretical analysis is a corroboration of the empirical relationship seems to be an exaggeration. In Liu et al’s reference 20 a much more measured discussion of the relevance of the Liu et al reference 39 to this power law relationship and the relevance of another paper by Wood is also noted (R. Wood, 2000: Parameterization of the effect of drizzle upon the droplet effective radius in stratocumulus clouds. Q. J. R. Meteorol. Soc. 126, 3309-3324.)

Reviewer #2

(Remarks to the Author)

This paper studies the cloud-mediated effect of anthropogenic aerosol emissions on Earth’s climate, known as Aerosol Cloud Interactions (ACI). Specifically, this paper aims to provide an observational quantification of the change in the shape/width of the cloud particle size distribution in response to changes in aerosol concentration, which influences how much solar radiation clouds scatter back to space and hence global temperatures. For this purpose, the authors employ satellite remote sensing observations.

I recommend rejection of this manuscript due to several significant methodological and conceptual issues detailed below. In combination, these issues leave me uncertain as to the efficacy of the methodology and the significance of the results.

Major Comments:

There is no uncertainty quantification. The error-covariance structure of V_e and R_e is necessary to know whether the observed co-variance between β and LWC/N_d is a real signal or a retrieval artifact. Even random errors in V_e will introduce a negative correlation between β and LWC/N_d as the two quantities are not independent. The authors have failed to quantify these effects and therefore exclude them as a cause for the observed relationships. Without this information, how can we believe that the Dispersion effect, as quantified here, is physical?

I emphasize that the dataset used, POLDER-NN, is a neural network that has been trained on synthetic data but has yet to be validated using in-situ observations. Without such error quantification, the data are not mature enough for use in a study such as this.

No attempt is made to establish the causality of the results. Instead, causality is implicitly assumed, and this distinction is not communicated to the reader. In particular, a co-varying relationship between β and LWC/N_d is found from the observations. Why should the cloud-response to anthropogenic aerosol emission patterns be constrained to follow this curve? How do we know this variability is not driven by other confounding factors?

The Dispersion effect is contained within the standard definitions of the Twomey effect and liquid water path adjustment (Bellouin et al., 2020). The authors have adopted non-standard definitions of the Twomey effect and liquid water path adjustment to present their results as if the Dispersion effect is “offsetting” the Twomey effect when in-reality it is simply part of the Twomey effect. Statements about the Dispersion effect offsetting the Twomey effect in the abstract and the manuscript are misleading as to the significance of the work.

The standard definition of the Twomey effect is the change in cloud albedo in response to a change in droplet number concentration for a fixed LWP. This includes the Dispersion effect, though I agree with the authors that it is not typically calculated separately or necessarily included in models.

As a result, this issue of semantics does not mean that the objective of this work is without significance or merit. By studying the Dispersion effect, we gain insight into the mechanisms controlling the Twomey effect, which can be useful for improving modelling of the Twomey effect and the interpretation of observational estimates. But, the Dispersion effect it is not entirely neglected. For example, an observational estimate of the Twomey effect which assumes that β is constant implicitly includes the Dispersion effect in their definition of droplet number concentration. What they are estimating is $\Delta\tau/\Delta(N_d \beta)$, but they are calling it $\Delta\tau/\Delta(N_d)$.

These distinctions need to be clearly and correctly communicated to the reader when arguing for the significance of the work and in the sequence of equations leading to Eq. 3 (Method 3). The step from M20 to M21, in particular, needs to be explained carefully. I would prefer a single derivation of the correct equations rather than two sets with one being incorrect.

The scale-dependence of the results is not adequately addressed. While the resolution of the analysis is 1 degree by 1 degree, this does not mean that the relationships are appropriate for a GCM just because a GCM is also coarse resolution. The co-variance of microphysical quantities at the sub-grid-scale will also affect the impact of the Dispersion effect in a GCM.

How large is this effect relative to the correlations among grid averages? Without this information it is unclear whether this study provides a useful quantification of the Dispersion effect.

The paper presents the results as being global but has significant sampling biases. Data is only included from clouds that are overcast at roughly ~6 km resolution. This excludes much of the boundary layer cloud coverage, which is typically composed of very small clouds. What does this mean for the global effect of the Dispersion effect? How would these results be used to provide a parameterization of the Dispersion effect in GCMs?

Other Comments:

The manuscript employs large numbers of symbols and quantities, many of which are extraneous. This leads to an overly lengthy presentation in the Methods, but especially in the first paragraph of the Main text. This first paragraph should be focused on rapidly introducing the necessary context for the wide readership of Nature Communications to understand and appreciate the work. A list of symbols and names without definitions to begin is not helpful. I suggest avoiding the term "spectral" entirely.

In the Methods, Eq. M2-M8 and associated text are extraneous. Eq. M9 can be stated to relate V_e to Beta based on Grosvenor et al., 2018, for example.

There is redundancy between paragraph 2 (Line 40) and the "Comparison with previous studies" section in describing previous results. I suggest leaving the details to the later section and providing just the range of values early on to motivate your work.

Version 1:

Reviewer comments:

Reviewer #2

(Remarks to the Author)

The authors have largely addressed my comments from the previous review. I commend them for their hard work.

As the authors acknowledge, the extension of the conclusions to cumuliform clouds is tenuous, due to instrumental limitations and lack of retrieval uncertainty quantification in these cloud types. However, the global impact of the Dispersion effect is dominated by the horizontally extensive clouds. Currently, this study reports a global impact of the Dispersion effect which includes uncertainty in the contribution from smaller or cumuliform clouds. I think it would be valuable to also quantify the Dispersion effect explicitly restricted to stratiform cloud types or only averaged over regions of the globe which are dominated by stratiform clouds. While less general, this quantity will be more conclusive, helping readers to interpret the study.

I think, with this clarification, my concerns will have been fully addressed.

(Remarks on code availability)

**Response to Referee #1:**

Thanks for the careful review and instructive comments. We have revised the paper
carefully based on your comments. The responses are described below (*italic text in*
*blue color is the reviewer's comment*).

**General Comments:**

*The noteworthy result in this paper is a global, observation-based quantitative*
*estimation of the impact of the dispersion effect (DE) on Aerosol-Cloud Interactions*
*(ACI), specifically the impact of the DE on changes to the planetary albedo that are*
*caused by droplet number concentration changes and liquid water path adjustments.*

*In particular, the way in which the authors formulate the dispersion effect as a*
*fractional offset to the radiative forcing caused by ACI allows it to be applied to a wide*
*range of existing results on ACI. The new dataset from POLDER, that provides both*
*effective radius and effective variance, offers the first set of global observations that*
*can be used to constrain the DE and the paper presents the benefits of this new global*
*dataset and the potential limitations caused by the spatial scale at which it is available*
*clearly. There are no obvious flaws in the analysis, interpretation and conclusions and*
*specific suggestions and corrections to errors, or the identification of conflicting*
*statements, are given below. The methods used are adequately presented in the Methods*
*section, but the clarity of presentation in the body of the paper leaves something to be*
*desired and suggestions for improvement are given below.*

*One of the most interesting figures to this reviewer is Fig. 2, and in particular the*
*difference between Fig 2e) and 2f). The empirical power law appears to be a much*
*better fit to the oceanic data than to the data over land (15% more variance explained).*
*It would be of interest to hear whether the authors have a good explanation for this.*

**Response:**

We sincerely appreciate the reviewer's positive evaluation and comments. The
reviewer acknowledged the value of our work and raised an insightful question
regarding the land-ocean differences in fitting performance.

Fig R1. The joint histograms of β_p versus $(LWC/N_d)_p$ in global ($60^\circ\text{S}\sim 60^\circ\text{N}$), land, and
 ocean regions, with logarithmic scales on both axes. (from Fig .2 in main text)

In this study, we consider LWC/N_d to be the dominant factor influencing β , and their
 relationship can be primarily represented by a power-law relationship. However, when
 LWC/N_d is either small or large, the relationship between β and LWC/N_d appears
 to deviate from this power-law, particularly in land areas (Fig. R1e). In other words,
 when LWC/N_d reaches extreme values, β may be significantly affected by other
 factors. We speculate that this is related to the following three factors:

- 1) The uncertainty of the retrieval algorithm: Compared to the situation over ocean,
 the retrieval uncertainty for large cloud droplets over land is larger (Fig R2, red
 box). Di Noia et al. (2019) revealed that these deviations mainly occur in
 southeastern China, around the Río de la Plata estuary in South America, and along
 the southern part of the Bight of Biafra in Africa (Di Noia et al., 2019, Fig. 14).
 They speculated that this may be related to the influence of aerosols above clouds
 in these regions on the POLDER-NNs method. This may cause β over land to
 become insensitive to LWC/N_d when $LWC/N_d > 5\text{ng}$, thereby reducing the
 fitted correlation coefficient (Fig. R1).

Fig R2. Global comparison between neural-network-based POLDER-3 effective radius
 retrieval (r_{eff} , i.e. R_e in our study), and MODIS retrievals over ocean (a) and land (b),
 encompassing the full year 2006 on a $1^\circ \times 1^\circ$ spatial grid. Effective radius values are in μm .
 (from Di Noia et al. (2019), Fig .13)

- 2) Condensation (evaporation) occurring simultaneously with new activation
 (deactivation) for small droplets: Lu et al. (2020) and Zhang et al. (2025)

demonstrated through in-situ observations and numerical simulations that ε , which
is proportional to β , shows a positive correlation with LWC/N_d for small cloud
droplets. This may deviate the $\beta - LWC/N_d$ relationship away from a negative
correlation. Compared to oceanic clouds, cloud droplets over land tend to be
smaller overall, which may contribute to the different fitting performance in land
and ocean.

3) The collision-coalescence process for large droplets: When LWC/N_d is large
enough, the collection process or precipitation may occur (Lu et al., 2012), thereby
disrupting the original characteristics of β . Considering that the retrieval of large
cloud droplets over land is inherently less accurate, this effect may be further
exacerbated.

The above are our hypotheses regarding the causes of the land-ocean differences, which
require further in-depth research for validation and have been added to the main text of
the paper. (Line 496~519 in the revised paper, similarly hereinafter)

***Detailed Comments:***

1) *The parameter beta is introduced at line 55, but is not defined. Given that this*
*parameter is used in many different ways it should at least be described as the ratio*
*of effective radius to volume mean radius, even if an equation is not given.*

Thank for your suggestion. We have rewritten it (Line: 57).

2) *The empirical equation linking beta with the ratio of Liquid Water Content (LWC)*
*to droplet number concentration (Nd) is given in the text at line 61, but is not*
*numbered, and cannot therefore be referenced. The order of the three steps*
*proposed to quantify the impact of DE on ACI (line 71) makes no sense. After beta*
*and (LWC/Nd) are estimated in step 1 the next step is surely to estimate a and b*
*using equation (M4). Equation (M4) should therefore be listed here in the Main*
*text so that it can be easily referenced as it is a key part of the analysis. Step 3*
*should therefore become Step 2 and can be properly described, rather than have*
*the current vague one line description. In the current Step 2 (which should become*
*Step 3) an equation is given providing the dependence of cloud optical depth on*
*cloud top variables and then referring to Method 3. I strongly suggest also noting*
*the form of the proportionality to LWP and reference eq.(17) of Reference 53.*
*Reference 53 provides one of the earliest and most detailed rationales for the use*
*of an adiabatically stratified framework in analyses of this type and making the*
*link to this reference early in the presentation would be helpful to the reader. It*
*also lists the constants of proportionality including the dependence on the*
*adiabatic liquid water lapse rate.*

Thanks a lot for your instructive suggestion. We have made adjustments and

revisions based on your suggestion to make the background narrative more logical.
(Lines: 64, 75~88, 383~389)

3) *In Eq.(3) it would be helpful to the reader if rather than using $DO_LWP=0.6*100\%$
it was written as $DO_LWP=3/5*100\%$, so that the interested reader can easily
make the link to the power laws given for the dependence of cloud optical depth on
cloud top variables.*

Thank you for your suggestion. This indeed makes it easy to link to the power-law
relationship, and the changes have been made accordingly. (Equations (3), (M24),
(M25), and (M27))

4) *The phrase “different grids” is used in several places (e.g. lines 101, 117). If you
mean “different grid points”, or “different spatial locations” please say so.*

Thank you for your correction. Our intention was to express "different grid points,"
and the necessary revision has been made. (Lines: 99, 101, 118, 269, 332, 454, 464)

5) *On lines 139 and 140 you state that “for the ocean (5.2 ng) is lower than that for
land (7.3 ng)”. The exact opposite is the case shown in Figs. 2b) and 2c). I believe
the statement in the text at lines 139/140 is wrong. Please correct text, or figure
appropriately.*

Thank you for pointing this out. This was a mistake in our text, and it has been
corrected. (Lines: 139~140)

6) *At lines 147/148 you state that “the absolute bias between the two methods for
global, land, and ocean regions are 0.04, 0.02, and 0.05, respectively”. Examining
Fig. 2 suggests these values are all too large by an order of magnitude. Please
correct.*

Thank you again for your careful reading and pointing these out. The corrections
have been made (Lines: 141, 146~148)

7) *In the discussion of adiabatic conditions, it is probably worth referring back to the
dependence of optical depth on adiabatic liquid water lapse rate given in eq.(17)
of Ref 53 which is $Cw^{(-1/6)}$. This suggests that variations in how sub-adiabatic a
profile is, as a result of entrainment mixing, is not going to have a substantial
impact on the albedo, or the results presented in this paper.*

Thank you for your suggestion. We have made revisions to the relevant content in
the Discussion section, which has been very helpful in enhancing the reliability of

our results. (Lines: 209~214).

8) *At line 202 the statement is made that “This empirical finding is corroborated by*
*the theoretical analysis conducted by Liu et al.39”. That Liu et al. paper makes the*
*link between dispersion and number concentration, but to say that theoretical*
*analysis is a corroboration of the empirical relationship seems to be an*
*exaggeration. In Liu et al’s reference 20 a much more measured discussion of the*
*relevance of the Liu et al reference 39 to this power law relationship and the*
*relevance of another paper by Wood is also noted (R. Wood, 2000:*
*Parameterization of the effect of drizzle upon the droplet effective radius in*
*stratocumulus clouds. Q. J. R. Meteorol. Soc. 126, 3309-3324.)*

Thank you for your suggestion. We have revised the relevant sentences to "The
empirical relationship between the two was demonstrated by Wood (2000), which
was later supported by the theoretical analysis of Liu et al. (2006) and further
validated by aircraft observations from several campaigns (Liu et al., 2008)."
(Lines: 219~222)

**Response to Referee #2:**

We are grateful for the careful review and instructive comments. We have revised the
paper carefully based on your comments. A point-by-point reply to the comments is
described as follows (*italic text in blue color is your comment*).

**General Comments:**

*This paper studies the cloud-mediated effect of anthropogenic aerosol emissions on*
*Earth's climate, known as Aerosol Cloud Interactions (ACI). Specifically, this paper*
*aims to provide an observational quantification of the change in the shape/width of the*
*cloud particle size distribution in response to changes in aerosol concentration, which*
*influences how much solar radiation clouds scatter back to space and hence global*
*temperatures. For this purpose, the authors employ satellite remote sensing*
*observations.*

*I recommend rejection of this manuscript due to several significant methodological and*
*conceptual issues detailed below. In combination, these issues leave me uncertain as to*
*the efficacy of the methodology and the significance of the results.*

**Response:**

We thank the reviewer for the positive confirmation of the main goal of our work. At
the same time, we would also like to express our sincere gratitude for the many
insightful and constructive comments you provided, which helps improve the quality
of our paper. These comments primarily addressed: 1) uncertainty quantification, 2)
causality analysis, and 3) potential scale-dependence and sampling bias inherent in our
method. Following your suggestions, we have employed new methods and
collected/analyzed a new observational dataset in the revision. We hope the revised
manuscript addresses your concerns.

**Major Comments:**

*1) There is no uncertainty quantification. The error-covariance structure of V_e and*
*R_e is necessary to know whether the observed co-variance between β and*
*LWC/N_d is a real signal or a retrieval artifact. Even random errors in V_e will*
*introduce a negative correlation between β and LWC/N_d as the two quantities are*
*not independent. The authors have failed to quantify these effects and therefore*
*exclude them as a cause for the observed relationships. Without this information,*
*how can we believe that the Dispersion effect, as quantified here, is physical?*

Thank you for your suggestion; this is indeed a crucial point, and we have

conducted an uncertainty analysis based on your recommendation.

Currently, the quantifiable sources of uncertainty include: 1) the inherent
limitations of the POLDER-NNs (Di Noia et al., 2019); 2) cloud heterogeneity
(Shang et al. 2015); 3) the retrieval method, wavelength, and grid scale (Shang et
al., 2019); and 4) the fitting method for parameter b (Jia and Quaas, 2023). Based
on previous studies, we determined the uncertainty range of b caused by different
sources and used the Monte Carlo method to determine the best estimate of b and
its 5 ~ 95% confidence interval for ocean, land, and global scales. The uncertainty
quantification process is shown in Fig. R3a. However, there are still some sources
of uncertainty that cannot be quantified currently, such as the effects of
entrainment-mixing and power-law relationship. These non-quantifiable
uncertainties are discussed in the Discussion section. The above analysis further
strengthens the robustness of our conclusions while providing readers with a more
quantitative understanding of the uncertainties in the current results. Next, the
uncertainty quantification process will be described in detail.

The first source of uncertainty (Src1) arises from the inherent uncertainty in the
POLDER-NNs method. The POLDER-NNs dataset employed in this study is
generated via a neural network algorithm, and discrepancies naturally exist
between this dataset and the exact solution derived from the radiative transfer
model. These discrepancies introduce uncertainty in the retrievals of R_e and V_e .
Based on Table 3 from Di Noia et al. (2019), Src1 is found to cause a bias ($Bias$)
and root mean square error ($RMSE$) in R_e/V_e of 0.08/-0.01 and 0.92/0.03,
respectively, as shown in Table R1. Based on this, we introduce a bias-corrected
random Gaussian noise for each R_e/V_e value in POLDER-NNs, denoted as
$N(-Bias, RMSE)$, where N represents a normal distribution with a mean of
$-Bias$ and a standard deviation of $RMSE^{1/2}$. Using the corrected values, we
calculate β and LWC/N_d , then fit the data to obtain the parameter b and its
corresponding standard error (SE) while accounting for the impact of Src1. To
ensure the robustness of the results and avoid potential biases from a single random
sampling, we repeat this process 10,000 times. The final estimates of b and SE ,
considering Src1, are obtained by averaging all sampled results and are denoted as
b_{s1_d} and SE_{s1_d} , where d indicates the use of the direct fitting method (Table
R1).

(a) Flowchart of uncertainty quantification

(b) Probability distribution function of b

Fig. R3: (a) The flowchart of uncertainty quantification, where b_{s_F} and SE_{s_F} represent the fitting parameter b and its standard error (SE) considering different sources of uncertainty ($S = s1, s2, s3$, representing Src1 to Src3) and using different fitting methods ($F = p, d$, representing pre-binned and direct fitting methods). $N(b_{s_F}, SE_{s_F})$ represents a normal distribution with a mean of b_{s_F} and a standard deviation of SE_{s_F} . (b) The probability distribution functions of parameter b in ocean, land, and global scales, where the point and errorbar represent the best estimate (i.e., the mean value) and its 5~95% confidence interval.

Table R1: Impacts of different sources of uncertainty on b . *Bias* and *RMSE* represent the
 deviation and root mean square error introduced by different sources in the retrieval of R_e and
 V_e . Variations in sources and fitting methods introduce uncertainty into the calculation of
 parameter b , denoted as b_{S_F} and SE_{S_F} , where S represents different sources, and F
 indicates the fitting method, with d for the direct method and p for the pre-binned method.
 These values correspond to the estimated b considering each source of uncertainty and its
 associated standard error (*SE*). The final row presents the ensemble uncertainty considering all
 sources, providing the best estimate and the 5~95% confidence interval.

Source and method	R_e	V_e	Fitting method	
			Direct	Pre-binned
Src1: Limitations of the POLDER-NNs	Bias = 0.08 RMSE = 0.92	Bias = -0.01 RMSE = 0.03	$b_{s1,d} = -0.026$ $SE_{s1,d} = 0.00004$	$b_{s1,p} = -0.021$ $SE_{s1,p} = 0.0014$
Src2: Heterogeneity of clouds	Bias = -0.71 RMSE = 0.88	Bias = 0.02 RMSE = 0.04	$b_{s2,d} = -0.029$ $SE_{s2,d} = 0.00005$	$b_{s2,p} = -0.024$ $SE_{s2,p} = 0.0014$
Src3: Retrieval method, wavelength, grid scale	/	/	$b_{s3,d} = -0.022$ $SE_{s3,d} = 0.0002$	$b_{s3,p} = -0.021$ $SE_{s3,p} = 0.007$
Ensemble via a Monte Carlo method	/	/	$b = -0.024 [-0.026 \sim -0.022]$	

The second source of uncertainty (Src2) stems from the impact of cloud
 heterogeneity. The POLDER-NNs data utilized in this study are characterized by a
 relatively coarse resolution, which may introduce errors by assuming homogeneity
 within the retrieval area (~6 km resolution). Therefore, it is essential to account for
 the effects of cloud heterogeneity on the retrievals of R_e and V_e . Shang et al.
 (2015) evaluated this impact by modeling a cloud field comprising several equal-
 area subregions with constant cloud optical thickness but varying R_e and V_e
 values. Based on the differences between the actual and retrieved R_e/V_e (as shown
 in Table 2 in Shang et al. (2015)), Src2 is found to cause the *Bias* and *RMSE* in
 R_e/V_e of -0.71/0.02 and 0.88/0.04, respectively (Table R1). Following the
 uncertainty quantification framework from Src1, the uncertainty in b caused by
 Src2 is then determined, and the $b_{s2,d}$ and $SE_{s2,d}$ are provided, as shown in
 Table R1.

The third source of uncertainty (Src3) arises from the use of the POLDER dataset,
 specifically related to the retrieval method, wavelength selection, and grid-scale
 processing. The POLDER-NNs dataset is derived via applying a machine learning
 to the multi-angle, multi-wavelength polarimetric measurements with a grid scale

of $1^\circ \times 1^\circ$. Shang et al. (2019) introduced an enhanced primary cloudbow retrieval
(PCR) algorithm to estimate R_e and V_e from POLDER, creating a global
retrieval dataset for four months (February, May, August, and November 2008)
(denoted as POLDER-PCR). Unlike POLDER-NNs, POLDER-PCR employs
traditional retrieval methods, clearly differentiating between various retrieval
wavelengths (670/865 nm), with a grid resolution of $0.7^\circ \times 0.7^\circ$. Data from
POLDER-PCR for low- and mid-latitude regions ($60^\circ\text{S} \sim 60^\circ\text{N}$) were selected for
fitting parameter b . Based on the fitting outcomes across two wavelengths, the
uncertainty in b due to Src3 is determined, and the corresponding b and SE are
provided (denoted as $b_{s3,d}$ and $SE_{s3,d}$), as shown in Table R1. Although
POLDER-NNs and POLDER-PCR utilize different wavelength and grid scale
configurations, as well as distinct retrieval methodologies, the derived b values
are relatively consistent. Compared to POLDER-NNs, the b values obtained from
POLDER-PCR exhibit larger SE , likely due to the smaller sample size of the
POLDER-PCR dataset (68,555 valid samples) and discontinuous V_e values
(intervals of 0.02). However, this convergence in b values obtained from two
independent retrievals provides a basis for evaluating the potential influence of
Src3.

The fourth source of uncertainty (Src4) arises from the fitting method used for
parameter b . As discussed in this paper (Lines 142~156), commonly used fitting
methods for large satellite datasets include direct and pre-binned fittings, though
the superior method remains unclear. Therefore, two fitting methods were
employed to derive b and SE using Src1~Src3, as shown in Table R1.

Finally, the overall impact of four sources on the estimation of b is quantified
using a Monte Carlo method, similar to that of Boucher and Haywood (2001) and
Bellouin et al. (2020). It is assumed that the b values induced by different sources,
as shown in Table R1, follow a normal distribution (i.e., $b \sim N(b_{S_F}, SE_{S_F}^2)$,
where S represents different sources, and F indicates the fitting method). A
random sampling process is then performed 10 million times, and for each random
result, the mean value of b affected by different sources is calculated, resulting in
a probability density distribution of b . The mean is then computed as the best
estimate of b (-0.024), and the 5~95% confidence intervals are determined as the
uncertainty range for b (-0.026 ~ -0.022), as shown in Fig. R3b and Table R1.

Similar uncertainty quantifications have also been applied to clouds over ocean and

land respectively, with the relevant parameters listed in Table R2 and the
 probability density distributions shown in Fig. R3b. Compared to the ocean region,
 the uncertainty in cloud results over the land is somewhat higher, mainly due to
 greater fitting uncertainty of the POLDER data in the oceanic region (Table R2).
 We suspect that this is related to three factors, including 1) the retrieval uncertainty
 for large cloud droplets over land is larger; 2) condensation (evaporation) occurring
 simultaneously with significant new activation (deactivation) for small droplets; 3)
 the collision-coalescence process for large droplets. Please see Lines 496~519 in
 the revised manuscript for more details.

 The above uncertainty quantitative analysis has been added to the main text in the
 "Quantitative Analysis of Uncertainty" section and to the Methods section under
 "Process of uncertainty quantification" (Line 157 ~ 176 and 520~581 in the revised
 paper, similarly hereinafter).

Table R2: Similar to Table R1 but for land and ocean scales.

Source	Land		Ocean	
	Direct	Pre-binned	Direct	Pre-binned
Src1	$b_{s1,d} = -0.026$	$b_{s1,p} = -0.023$	$b_{s1,d} = -0.030$	$b_{s1,p} = -0.025$
	$SE_{s1,d} = 0.00006$	$SE_{s1,p} = 0.0024$	$SE_{s1,d} = 0.00007$	$SE_{s1,p} = 0.0013$
Src2	$b_{s2,d} = -0.029$	$b_{s2,p} = -0.025$	$b_{s2,d} = -0.034$	$b_{s2,p} = -0.028$
	$SE_{s2,d} = 0.00007$	$SE_{s2,p} = 0.0027$	$SE_{s2,d} = 0.00008$	$SE_{s2,p} = 0.0014$
Src3	$b_{s3,d} = -0.023$	$b_{s3,p} = -0.023$	$b_{s3,d} = -0.030$	$b_{s3,p} = -0.025$
	$SE_{s3,d} = 0.0006$	$SE_{s3,p} = 0.010$	$SE_{s3,d} = 0.0003$	$SE_{s3,p} = 0.008$
Ensemble	$b = -0.025 [-0.028 \sim -0.022]$		$b = -0.029 [-0.031 \sim -0.026]$	

 2) *I emphasize that the dataset used, POLDER-NN, is a neural network that has been*
 *trained on synthetic data but has yet to be validated using in-situ observations.*
 *Without such error quantification, the data are not mature enough for use in a*
 *study such as this.*

Thank you for your comment. You are correct that conducting in-situ validation is
 an important approach to ensure the reliability of the data. However, limited,
 existing in-situ aircraft observations and POLDER observations do not perfectly
 match in terms of cloud height (aircraft data were mostly obtained within the cloud,
 while the satellite data were taken above the cloud top), making direct comparison
 and validation difficult and very limited. Furthermore, The POLDER data we used

comes from observations onboard an earth-observing satellite in polar orbit, which
adds further challenges in finding time-matching aircraft observation data. Based
on our investigation, there are currently no available in-situ observations
corresponding to POLDER-NNs data, which is the main reason why this study did
not conduct such validation. However, POLDER-NNs data have been compared
with MODIS results and have shown good consistency (Di Noia et al., 2019). In
turn, the quality and limitations of the MODIS effective radius retrievals are fairly
well investigated with in situ observations (Grosvenor et al., 2018; Painemal and
Zuidema, 2011; Witte et al., 2018). Furthermore, the β derived and calculated
through V_e is also within the valid range of previous observational and model
calculation results (Liu et al., 2008; Wang et al., 2020). Thus we believe the
conclusions drawn in this study are trustworthy. Nevertheless, as you pointed out,
it is necessary for readers to be aware of the limitations of this study, so we have
included the relevant discussion in Lines: 249~251.

*3) No attempt is made to establish the causality of the results. Instead, causality is*
*implicitly assumed, and this distinction is not communicated to the reader. In*
*particular, a co-varying relationship between β and LWC/N_d is found from the*
*observations. Why should the cloud-response to anthropogenic aerosol emission*
*patterns be constrained to follow this curve? How do we know this variability is*
*not driven by other confounding factors?*

Thank you for your comment. The previous content was indeed insufficient, so we
have added more details to explain the rationale of this causal assumption from
both theoretical derivation and statistical analysis perspectives.

1) First, this causal relationship aligns with our physical understanding of cloud
microphysical processes. The formation of cloud droplet size spectra is primarily
controlled by the condensation growth process (Liu et al., 2008; Liu and Li, 2015).
And the negative relationship basically reflects the fact that condensation leads to
a narrow size distribution as droplets grow. The empirical relationship between β
and LWC/N_d was demonstrated by Wood (2000), which was later supported by
the theoretical analysis of Liu et al. (2006) and further validated by aircraft
observations from several campaigns (Liu et al., 2008). Therefore, we think that
the co-varying relationship between β and LWC/N_d has solid physics behind it,
instead of being an observational artifact.

2) In addition, to address your concern, we have conducted causal analysis, which

further confirms the conclusions. To examine the relationship between aerosols,
 LWC/N_d , and β , as well as the mediating effect of LWC/N_d on the impact of
 aerosols on β , we use the POLDER Level 3 aerosol products, generated using
 Generalized Retrieval of Atmosphere and Surface Properties "components"
 approach, gridded at a $1^\circ \times 1^\circ$ resolution (POLDER-3/GRASP, version 1.1)
 (Dubovik et al., 2011, 2014, 2021; Li et al., 2019). This product is officially
 recommended for studies involving both the Ångström Exponent (AE) and Aerosol
 Optical Depth (AOD) and demonstrates a high consistency with the Aerosol
 Robotic Network (AERONET) on a global scale (Chen et al., 2020; Li et al., 2019).

To better characterize the properties of aerosols that can be activated as cloud
 droplets, this study uses AE and AOD at 565 nm from the POLDER-3/GRASP
 to calculate the aerosol index (AI , $AI = AE \times AOD$) (Jia and Quaas, 2023). Before
 analysis, AI were matched with POLDER-NNs on a daily basis within a $1^\circ \times 1^\circ$
 grid to ensure full spatiotemporal alignment between the two datasets. The results
 of the matched data analysis are shown in Fig. R4, where the power-law fitting of
 LWC/N_d and β with AI is performed using the pre-binned fitting, and the
 mediation effect analysis is conducted using the R package for a Causal Mediation
 Analysis (Tingley et al., 2014).

Fig. R4: (a-b) Same as Fig. R1 but for $(LWC/N_d)_P$ and β_P versus aerosol index (AI). (c)
 The result of mediation analysis, where ACMI and ADI represent Average Causal Mediation
 Impact and Average Direct Impact, respectively.

As shown in Figs. R4a, b, AI and LWC/N_d exhibit a negative power-law
 relationship, whereas AI and β show an overall positive power-law correlation,
 which is consistent with theoretical analysis. When aerosol concentration increases
 and the LWC in the cloud remains relatively stable, the liquid water per cloud
 droplet (LWC/N_d) decreases. At the same time, β increases overall with AI ,
 aligning with most previous studies that reported aerosol-induced broadening of
 the cloud droplet spectrum based on aircraft measurements (Liu et al., 2008; Liu

and Daum, 2002; Martin et al., 1994; Peng and Lohmann, 2003). Given the
negative power-law relationship between LWC/N_d and β , it is proved that
LWC/N_d plays a crucial mediating role in the impact of AI on β .

To investigate this, we have conducted a Causal Mediation Analysis to examine
the role of LWC/N_d as a mediator in AI 's impact on β , with the results shown
in Fig. R4c. The y-axis represents the change in $\ln(\beta)$ due to a one-unit increase
in $\ln(AI)$. The ACMI (Average Causal Mediation Impact) indicates the impact of
$\ln(AI)$ on $\ln(\beta)$ transmitted through the mediator $\ln(LWC/N_d)$ (0.0037),
while the ADI (Average Direct Impact) represents the direct impact of $\ln(AI)$ on
$\ln(\beta)$ (0.0041). The total impact is 0.0078, and all results are statistically
significant ($p < 0.001$). The above result suggests that

① ACMI and ADI are close in magnitude: the relationship between LWC/N_d
and β captures most of AI 's effect on β , confirming the critical mediating role of
LWC/N_d ;

② ACMI is slightly lower than ADI: changes in AI primarily reflect the effect of
N_d on β and the inclusion of LWC as a factor (i.e., LWC/N_d) partially offsets
the effect of N_d , which aligns with the discussion on DO_{LWP} in the main text.

In summary, we think that the co-varying relationship between β and LWC/N_d
largely reflects the changes in β induced by AI through its influence on
LWC/N_d . Therefore, the cloud response to anthropogenic aerosol emission
patterns is highly likely to follow this curve.

The above Causal analysis has been added to the main text in the "Causal analysis
of aerosol effects on LN and β " section and to the Methods section under "Causal
mediation analysis" (Lines: 215~241, 582~596).

4) *The Dispersion effect is contained within the standard definitions of the Twomey*
*effect and liquid water path adjustment (Bellouin et al., 2020). The authors have*
*adopted non-standard definitions of the Twomey effect and liquid water path*
*adjustment to present their results as if the Dispersion effect is "offsetting" the*
*Twomey effect when in-reality it is simply part of the Twomey effect. Statements*
*about the Dispersion effect offsetting the Twomey effect in the abstract and the*
*manuscript are misleading as to the significance of the work. The standard*
*definition of the Twomey effect is the change in cloud albedo in response to a*
*change in droplet number concentration for a fixed LWP. This includes the*

*Dispersion effect, though I agree with the authors that it is not typically calculated*
*separately or necessarily included in models*

Thank you for your suggestion. As explained in the Main text of our paper (Lines:
40~42, 57~59), the dispersion effect (DE) does not refer to the absolute magnitude
of β but rather to the change in cloud albedo caused by the response of β to
aerosol variations. Bellouin et al. (2020) did not include DE in the definition of the
Twomey effect, as the relationship between τ and β was not considered in their
Eq. 11 (β was set as a constant, making its sensitivity to aerosols 0, effectively
excluding DE). The complete formulation that accounts for DE can be found in Eq.
12.12 of Quaas & Gryspeerdt (2022), which is also adopted in this study. However,
we agree with your description of the Twomey effect, which refers to the cloud
albedo effect under constant LWP (influenced by both N_d and β). Therefore, to
more clearly distinguish the individual impacts of N_d and β on the cloud albedo
effect, we will use “number effect” instead of the Twomey effect to clarify this
point. Accordingly, the definitions of F_{DE-N_d} and DO_{N_d} have also been
modified, as described in the revised version. (Lines: 43~44, 85~88)

5) *As a result, this issue of semantics does not mean that the objective of this work is*
*without significance or merit. By studying the Dispersion effect, we gain insight*
*into the mechanisms controlling the Twomey effect, which can be useful for*
*improving modelling of the Twomey effect and the interpretation of observational*
*estimates. But, the Dispersion effect it is not entirely neglected. For example, an*
*observational estimate of the Twomey effect which assumes that β is constant*
*implicitly includes the Dispersion effect in their definition of droplet number*
*concentration. What they are estimating is $\Delta\tau/\Delta(N_d \beta)$, but they are calling it*
*$\Delta\tau/\Delta(N_d)$.*

Thank you for your comments. We appreciate your first comment, which highlights
the significance of separating and studying the DE in this research. Regarding your
second comment, there may have been some misunderstanding of the concept of
the DE due to our previous unclear description. According to Eq. (12.49) in Quaas
& Gryspeerdt (2022), $\tau_c \propto LWP^{5/6}(kN_d)^{1/3}$, where $k = \beta^{-3}$. Therefore, we
think what you are referring to is $\Delta\tau/\Delta(kN_d)$ and $\Delta\tau/\Delta N_d$.

It is necessary to point out that kN_d and N_d are not equivalent. Firstly,
numerically, k is typically less than 1 in actual observations, meaning kN_d is
numerically lower than N_d . Secondly, in terms of physical meaning, the
appearance of kN_d arises from the need to convert R_v to R_e (Eq. (12.47-48) in

Quaas & Gryspeerdt (2022). Thus, $\Delta\tau/\Delta(kN_d)$ to some extent represents the
sensitivity of τ to R_e , while $\Delta\tau/\Delta N_d$ reflects the sensitivity of τ to R_v . Finally,
the traditional approach calculates $\Delta\tau/\Delta N_d$ rather than $\Delta\tau/\Delta(kN_d)$ because,
when β is constant, τ is not correlated with β , and thus $\tau_c \propto LWP^{5/6} N_d^{1/3}$.
Coupled with the definition of DE (the change in cloud albedo caused by aerosol
changes that lead to β variations, i.e., DE is a sensitivity, not the absolute value
of β), it is evident that previous studies have not implicitly considered DE in the
calculation of ACI.

6) *These distinctions need to be clearly and correctly communicated to the reader*
*when arguing for the significance of the work and in the sequence of equations*
*leading to Eq. 3 (Method 3). The step from M20 to M21, in particular, needs to be*
*explained carefully. I would prefer a single derivation of the correct equations*
*rather than two sets with one being incorrect.*

Thank you for your suggestion. We have included additional clarification in the
transition from M20 to M21 to assist readers in understanding the rationale behind
the development of M21 and its connection to M20. The added explanation is as
follows:

“Although M20 incorporates β in the definition of τ_c (M12), β is treated as a
fixed parameter during practical implementation. As a result, variations in β
cannot be accounted for when evaluating τ_c under anthropogenic aerosol
perturbations (M13), thereby neglecting the influence of the dispersion effect in the
estimation of ERF_{aci} (M14-15). When the dispersion effect is considered (i.e.,
when β is treated as a variable rather than a constant), τ_c can be expressed as: ...”

The corresponding discussion can be found in Lines: 413~417.

7) *The scale-dependence of the results is not adequately addressed. While the*
*resolution of the analysis is 1 degree by 1 degree, this does not mean that the*
*relationships are appropriate for a GCM just because a GCM is also coarse*
*resolution. The co-variance of microphysical quantities at the subgrid-scale will*
*also affect the impact of the Dispersion effect in a GCM. How large is this effect*
*relative to the correlations among grid averages? Without this information it is*
*unclear whether this study provides a useful quantification of the Dispersion effect.*

Thank you for your suggestion. We agree on the importance of scale-dependence.
The parameterization scheme inherently has grid dependency, and a
parameterization scheme that matches the grid is more suitable for model
applications. The relationship between β and N_d or LWC/N_d is supported not

only by cloud microphysical theory (Liu et al., 2006) and simulations based on the
parcel model (Chen et al., 2016), but also by previous aircraft observation studies
conducted at cloud or regional scales across multiple regions (Liu et al., 2008;
Wood, 2000). What has been lacking, however, is evidence from large-scale
datasets. Therefore, our work is not an isolated finding but rather a further
confirmation of earlier theoretical studies and limited aircraft-based observations
with large scale satellite observations—one that is particularly applicable to large-
scale model applications. Additionally, we are unable to obtain higher-resolution
POLDER-NNs data, which is also why we did not perform the related analysis in
this study. In the future, if satellite data with finer spatial resolution becomes
available (e.g., from the Multi-viewing Multi-channel Multi-polarization Imaging),
we look forward to conducting higher-resolution $\beta - LWC/N_d$ analyses and
performing validation through comparison with aircraft observations.

8) *The paper presents the results as being global but has significant sampling biases.*
*Data is only included from clouds that are overcast at roughly ~6 km resolution.*
*This excludes much of the boundary layer cloud coverage, which is typically*
*composed of very small clouds. What does this mean for the global effect of the*
*Dispersion effect? How would these results be used to provide a parameterization*
*of the Dispersion effect in GCMs?*

Thank you for your instructive comment, and we agree with your suggestions. The
scheme we developed is primarily designed for calculating β and DE in large-
scale stratus/stratocumulus clouds in GCMs. In fact, current data are difficult to
achieve the high resolution needed for investigating small clouds like shallow
cumuli. We are hopeful that future high-resolution satellite observations will
enable us to explore the impact of these smaller clouds on the results.

To address your suggestions, we have revised the writing to clearly state that the
paper focuses on large-scale clouds. Additionally, we have included a brief
paragraph discussing cumulus clouds as a direction for future research. (Lines:
243~245)

***Other Comments:***

9) *The manuscript employs large numbers of symbols and quantities, many of which*
*are extraneous. This leads to an overly lengthy presentation in the Methods, but*
*especially in the first paragraph of the Main text. This first paragraph should be*
*focused on rapidly introducing the necessary context for the wide readership of*
*Nature Communications to understand and appreciate the work. A list of symbols*

*and names without definitions to begin is not helpful. I suggest avoiding the term*
*“spectral” entirely.*

Thank you for your suggestion. We have revised the narrative in the Introduction
section, including the removal of the term “spectral” throughout and the
replacement of LWC/N_d with LN , among other adjustments. As you pointed out,
the background section should avoid excessive use of equations and symbols
whenever possible. However, given the complexity of our calculation method, we
aimed to concisely present the core computational approach within the background
to help readers better understand the subsequent results.

*10) In the Methods, Eq. M2-M8 and associated text are extraneous. Eq. M9 can be*
*stated to relate V_e to Beta based on Grosvenor et al., 2018, for example.*

Thank you for your suggestion. After considering the feedback from both reviewers,
we think it is necessary to retain the introduction of these basic definitions, as they
will help a broader audience understand the physical foundation of our work. It is
possible that the current presentation is not concise and clear enough, which may
have led to the perception that this section is not closely related to the main focus
of the article. We will revise the relevant narrative accordingly. (Lines 343~368)

*11) There is redundancy between paragraph 2 (Line 40) and the “Comparison with*
*previous studies” section in describing previous results. I suggest leaving the*
*details to the later section and providing just the range of values early on to*
*motivate your work.*

Thank you for your suggestion. We have revised the content accordingly. (Lines:
40~56)

**References**

- Bellouin, N., Quaas, J., Gryspeerdt, E., Kinne, S., Stier, P., Watson-Parris, D., Boucher,
O., Carslaw, K. S., Christensen, M., Daniau, A. -L., Dufresne, J. -L., Feingold, G.,
Fiedler, S., Forster, P., Gettelman, A., Haywood, J. M., Lohmann, U., Malavelle, F.,
Mauritsen, T., McCoy, D. T., Myhre, G., Mülmenstädt, J., Neubauer, D., Possner, A.,
Rugenstein, M., Sato, Y., Schulz, M., Schwartz, S. E., Sourdeval, O., Storelvmo, T.,
Toll, V., Winker, D., and Stevens, B.: Bounding Global Aerosol Radiative Forcing of
Climate Change, *Reviews of Geophysics*, 58, <https://doi.org/10.1029/2019RG000660>,
2020.
- Boucher, O. and Haywood, J.: On summing the components of radiative forcing of
climate change, *Clim Dyn*, 18, 297–302,
<https://doi.org/10.1007/S003820100185/METRICS>, 2001.
- Chen, C., Dubovik, O., Fuertes, D., Litvinov, P., Lapyonok, T., Lopatin, A., Ducos, F.,
Derimian, Y., Herman, M., Tanré, D., Remer, L. A., Lyapustin, A., Sayer, A. M., Levy,
R. C., Hsu, N. C., Descloitres, J., Li, L., Torres, B., Karol, Y., Herrera, M., Herreras,
587 M., Aspetsberger, M., Wanzelboeck, M., Bindreiter, L., Marth, D., Hangler, A., and
588 Federspiel, C.: Validation of GRASP algorithm product from POLDER/PARASOL
data and assessment of multi-angular polarimetry potential for aerosol monitoring,
*Earth Syst Sci Data*, 12, 3573–3620, <https://doi.org/10.5194/essd-12-3573-2020>, 2020.
- Chen, J., Liu, Y., Zhang, M., and Peng, Y.: New understanding and quantification of
the regime dependence of aerosol-cloud interaction for studying aerosol indirect effects,
*Geophys Res Lett*, 43, 1780–1787, <https://doi.org/10.1002/2016GL067683>, 2016.
- Dubovik, O., Herman, M., Holdak, A., Lapyonok, T., Tanré, D., Deuzé, J. L., Ducos,
F., Sinyuk, A., and Lopatin, A.: Statistically optimized inversion algorithm for
enhanced retrieval of aerosol properties from spectral multi-angle polarimetric satellite
observations, *Atmos Meas Tech*, 4, 975–1018, <https://doi.org/10.5194/amt-4-975-2011>,
2011.
- Dubovik, O., Lapyonok, T., Litvinov, P., Herman, M., Fuertes, D., Ducos, F., Torres,
B., Derimian, Y., Huang, X., Lopatin, A., Chaikovsky, A., Aspetsberger, M., and
Federspiel, C.: GRASP: a versatile algorithm for characterizing the atmosphere, *SPIE*
*Newsroom*, <https://doi.org/10.1117/2.1201408.005558>, 2014.
- Dubovik, O., Fuertes, D., Litvinov, P., Lopatin, A., Lapyonok, T., Dubovik, I., Xu, F.,
Ducos, F., Chen, C., Torres, B., Derimian, Y., Li, L., Herreras-Giralda, M., Herrera, M.,
Karol, Y., Matar, C., Schuster, G. L., Espinosa, R., Puthukkudy, A., Li, Z., Fischer, J.,
Preusker, R., Cuesta, J., Kreuter, A., Cede, A., Aspetsberger, M., Marth, D., Bindreiter,
607 L., Hangler, A., Lanzinger, V., Holter, C., and Federspiel, C.: A Comprehensive
Description of Multi-Term LSM for Applying Multiple a Priori Constraints in Problems
of Atmospheric Remote Sensing: GRASP Algorithm, Concept, and Applications,
*Frontiers in Remote Sensing*, 2, 706851, <https://doi.org/10.3389/frsen.2021.706851>,
2021.

Grosvenor, D. P., Sourdeval, O., Zuidema, P., Ackerman, A., Alexandrov, M. D.,
Bennartz, R., Boers, R., Cairns, B., Chiu, J. C., Christensen, M., Deneke, H., Diamond,
614 M., Feingold, G., Fridlind, A., Hünerbein, A., Knist, C., Kollias, P., Marshak, A.,
McCoy, D., Merk, D., Painemal, D., Rausch, J., Rosenfeld, D., Russchenberg, H.,
Seifert, P., Sinclair, K., Stier, P., van Diedenhoven, B., Wendisch, M., Werner, F.,
Wood, R., Zhang, Z., and Quaas, J.: Remote Sensing of Droplet Number Concentration
in Warm Clouds: A Review of the Current State of Knowledge and Perspectives,
*Reviews of Geophysics*, 56, 409–453, <https://doi.org/10.1029/2017RG000593>, 2018.

Jia, H. and Quaas, J.: Nonlinearity of the cloud response postpones climate penalty of
mitigating air pollution in polluted regions, *Nat Clim Chang*, 13, 943–950,
<https://doi.org/10.1038/s41558-023-01775-5>, 2023.

Li, L., Dubovik, O., Derimian, Y., Schuster, G. L., Lapyonok, T., Litvinov, P., Ducos,
F., Fuertes, D., Chen, C., Li, Z., Lopatin, A., Torres, B., and Che, H.: Retrieval of
aerosol components directly from satellite and ground-based measurements, *Atmos*
*Chem Phys*, 19, 13409–13443, <https://doi.org/10.5194/acp-19-13409-2019>, 2019.

Liu, Y. and Daum, P. H.: Anthropogenic aerosols: Indirect warming effect from
dispersion forcing, *Nature*, 419, 580–581, <https://doi.org/10.1038/419580a>, 2002.

Liu, Y. and Li, W. L.: A method for solving relative dispersion of the cloud droplet
spectra, *Sci China Earth Sci*, 58, 929–938, <https://doi.org/10.1007/s11430-015-5059-9>,
2015.

Liu, Y., Daum, P. H., and Yum, S. S.: Analytical expression for the relative dispersion
of the cloud droplet size distribution, *Geophys Res Lett*, 33, 2810,
<https://doi.org/10.1029/2005GL024052>, 2006.

Liu, Y., Daum, P. H., Guo, H., and Peng, Y.: Dispersion bias, dispersion effect, and the
aerosol-cloud conundrum, *Environmental Research Letters*, 3, 045021,
<https://doi.org/10.1088/1748-9326/3/4/045021>, 2008.

Lu, C., Liu, Y., Niu, S., and Vogelmann, A. M.: Observed impacts of vertical velocity
on cloud microphysics and implications for aerosol indirect effects, *Geophys Res Lett*,
39, <https://doi.org/10.1029/2012GL053599>, 2012.

Lu, C., Liu, Y., Yum, S. S., Chen, J., Zhu, L., Gao, S., Yin, Y., Jia, X., and Wang, Y.:
Reconciling Contrasting Relationships Between Relative Dispersion and Volume-
Mean Radius of Cloud Droplet Size Distributions, *Journal of Geophysical Research:*
*Atmospheres*, 125, <https://doi.org/10.1029/2019JD031868>, 2020.

Martin, G. M., Johnson, D. W., and Spice, A.: The Measurement and Parameterization
of Effective Radius of Droplets in Warm Stratocumulus Clouds, *J Atmos Sci*, 51, 1823–
1842, [https://doi.org/10.1175/1520-0469\(1994\)051<1823:tmapoe>2.0.co;2](https://doi.org/10.1175/1520-0469(1994)051<1823:tmapoe>2.0.co;2), 1994.

Di Noia, A., Hasekamp, O. P., van Diedenhoven, B., and Zhang, Z.: Retrieval of liquid
water cloud properties from POLDER-3 measurements using a neural network
ensemble approach, *Atmos Meas Tech*, 12, 1697–1716, <https://doi.org/10.5194/amt-12-1697-2019>, 2019.

Painemal, D. and Zuidema, P.: Assessment of MODIS cloud effective radius and
optical thickness retrievals over the Southeast Pacific with VOCALS-REx in situ

measurements, *Journal of Geophysical Research: Atmospheres*, 116, n/a-n/a,
<https://doi.org/10.1029/2011JD016155>, 2011.

Peng, Y. and Lohmann, U.: Sensitivity study of the spectral dispersion of the cloud
droplet size distribution on the indirect aerosol effect, *Geophys Res Lett*, 30,
<https://doi.org/10.1029/2003GL017192>, 2003.

Quaas, J. and Gryspeerdt, E.: Aerosol-cloud interactions in liquid clouds, in: *Aerosols*
*and Climate*, vol. 13, Elsevier, 489–544, [https://doi.org/10.1016/B978-0-12-819766-](https://doi.org/10.1016/B978-0-12-819766-0.00019-5)
[0.00019-5](https://doi.org/10.1016/B978-0-12-819766-0.00019-5), 2022.

Shang, H., Chen, L., Bréon, F. M., Letu, H., Li, S., Wang, Z., and Su, L.: Impact of
cloud horizontal inhomogeneity and directional sampling on the retrieval of cloud
droplet size by the POLDER instrument, *Atmos Meas Tech*, 8, 4931–4945,
<https://doi.org/10.5194/amt-8-4931-2015>, 2015.

Shang, H., Letu, H., Bréon, F.-M., Riedi, J., Ma, R., Wang, Z., Nakajima, T. Y., Wang,
Z., and Chen, L.: An improved algorithm of cloud droplet size distribution from
POLDER polarized measurements, *Remote Sens Environ*, 228, 61–74,
<https://doi.org/10.1016/j.rse.2019.04.013>, 2019.

Tingley, D., Yamamoto, T., Hirose, K., Keele, L., and Imai, K.: mediation: R Package
for Causal Mediation Analysis, *J Stat Softw*, 59, 1–38,
<https://doi.org/10.18637/jss.v059.i05>, 2014.

Wang, M., Peng, Y., Liu, Y., Liu, Y., Xie, X., and Guo, Z.: Understanding Cloud
Droplet Spectral Dispersion Effect Using Empirical and Semi-Analytical
Parameterizations in NCAR CAM5.3, *Earth and Space Science*, 7, e2020EA001276,
<https://doi.org/10.1029/2020EA001276>, 2020.

Witte, M. K., Yuan, T., Chuang, P. Y., Platnick, S., Meyer, K. G., Wind, G., and
Jonsson, H. H.: MODIS Retrievals of Cloud Effective Radius in Marine Stratocumulus
Exhibit No Significant Bias, *Geophys Res Lett*, 45, 10,656–10,664,
<https://doi.org/10.1029/2018GL079325>, 2018.

Wood, R.: Parametrization of the effect of drizzle upon the droplet effective radius in
stratocumulus clouds, *Quarterly Journal of the Royal Meteorological Society*, 126,
3309–3324, <https://doi.org/10.1002/QJ.49712657015>, 2000.

Zhang, P., Wang, Y., Li, J., Fang, F., Zhu, L., and Lv, J.: Improved Parameterization
of Cloud Droplet Spectral Dispersion Expected to Reduce Uncertainty in Evaluating
Aerosol Indirect Effects, *Geophys Res Lett*, 52, e2024GL111643,
<https://doi.org/10.1029/2024GL111643>, 2025.

**Response to Referee #2:**

Thanks for the careful review and instructive comments. We have revised the paper
carefully based on your comments. The responses are described below (*italic text in*
*blue color is the reviewer's comment*).

**Comments:**

*The authors have largely addressed my comments from the previous review. I commend*
*them for their hard work.*

*As the authors acknowledge, the extension of the conclusions to cumuliform clouds is*
*tenuous, due to instrumental limitations and lack of retrieval uncertainty quantification*
*in these cloud types. However, the global impact of the Dispersion effect is dominated*
*by the horizontally extensive clouds. Currently, this study reports a global impact of*
*the Dispersion effect which includes uncertainty in the contribution from smaller or*
*cumuliform clouds. I think it would be valuable to also quantify the Dispersion effect*
*explicitly restricted to stratiform cloud types or only averaged over regions of the globe*
*which are dominated by stratiform clouds. While less general, this quantity will be more*
*conclusive, helping readers to interpret the study.*

*I think, with this clarification, my concerns will have been fully addressed.*

**Response:**

We sincerely appreciate the reviewer's positive assessment of our revised manuscript
and are grateful for the suggestion to clarify the cloud types considered in this study.

In the revised version, we provided detailed information about the classified cloud types
in the POLDER-NNs dataset, which is based on the cloud classification of the
International Satellite Cloud Climatology Project (ISCCP) (Fig. 1, Rossow et al., 2016),
to explicitly determine the cloud types targeted in this study. The classification results
are shown in Fig. 2, indicating that stratocumulus, altostratus, cumulus, altocumulus,
nimbostratus, and stratus account for 51.2%, 22.2%, 15.8%, 5.7%, 2.8%, and 2.4% of
the samples, respectively. Stratocumulus clouds, despite their partly cumulus-like
appearance, are typically categorized as stratiform clouds due to their broad horizontal
extent and limited vertical development (Wood, 2015). In addition, due to the strict
cloud cover criterion (cloud cover > 0.95 within the 5 km × 6 km grid), the cumuliform
clouds included here are limited to those with relatively large horizontal scales, while

small, isolated cumulus clouds (e.g., significantly smaller than 5 km × 6 km) are
excluded.

In summary, the analyses of cloud particle size distribution, the quantification of the
dispersion effect, and the development of the parameterization in this study are
primarily applicable to liquid-phase stratiform clouds (stratocumulus, altostratus,
nimbostratus, and stratus, collectively accounting for over 78% of the samples). While
some large-scale cumulus clouds are included in the dataset, the applicability of our
findings to typical cumulus clouds remains uncertain and warrants further investigation.
Additionally, this study specifically focuses on the impact of the dispersion effect on
aerosol-cloud interactions (ACI). Since cumulus clouds generally play a less dominant
role in ACI due to their short lifetimes, small spatial coverage, and weak coupling with
large-scale radiative processes (Fan et al., 2016), our focus on liquid-phase stratiform
clouds aligns well with the primary goals of current ACI research.

The relevant descriptions have been incorporated into the Abstract (Lines: 21~22),
Introduction (Lines: 76~77), Results (Lines: 92~93, 132~133, and 206~207),
Discussion (Lines: 245~253 and 266~267), and Methods (Lines: 342~358) of the
revised manuscript to explicitly clarify the applicability of our conclusions and
parameterization. The classification figure (Fig. 2) have been added to the
Supplementary Information as Supplementary Fig. 3.

**Fig. 1:** Definition of ISCCP cloud types based on Cloud Top Pressure and Cloud Optical
Thickness. (from ISCCP-H DATA USERS GUIDE)

**Fig. 2:** The number of samples and corresponding percentages of cloud types in the POLDER-
 NNs dataset used in this study are categorized following the ISCCP classification scheme
 (based on Cloud Top Pressure [p_t] and Cloud Optical Thickness [τ_c]). Specifically, Cu refers
 to cumulus ($680 \text{ hpa} < p_t \leq 1025 \text{ hpa}$, $0 < \tau_c \leq 3.55$), Sc to stratocumulus ($680 \text{ hpa} <$
 $p_t \leq 1025 \text{ hpa}$, $3.55 < \tau_c \leq 22.63$), St to stratus ($680 \text{ hpa} < p_t \leq 1025 \text{ hpa}$, $22.63 <$
 $\tau_c \leq 450$), Ac to altocumulus ($440 \text{ hpa} < p_t \leq 680 \text{ hpa}$, $0 < \tau_c \leq 3.55$), As to altostratus
 ($440 \text{ hpa} < p_t \leq 680 \text{ hpa}$, $3.55 < \tau_c \leq 22.63$), and Ns to nimbostratus ($440 \text{ hpa} < p_t \leq$
 680 hpa , $22.63 < \tau_c \leq 450$).

**References:**

Fan, J., Wang, Y., Rosenfeld, D., and Liu, X.: Review of Aerosol–Cloud Interactions:
Mechanisms, Significance, and Challenges, *J Atmos Sci*, 73, 4221–4252,
<https://doi.org/10.1175/JAS-D-16-0037.1>, 2016.

Rossow, W., Golea, V., Walker, A., Knapp, K., Young, A., Inamdar, A., Hankins, B.,
and NOAA’s Climate Data Record Program: International Satellite Cloud Climatology
Project Climate Data Record, H-Series v1.00 NOAA National Centers for
Environmental Information, <https://doi.org/https://doi.org/10.7289/V5QZ281S>, 2016.

Wood, R.: CLOUDS AND FOG | Stratus and Stratocumulus, in: *Encyclopedia of*
*Atmospheric Sciences*, Elsevier, 196–200, [https://doi.org/10.1016/B978-0-12-382225-](https://doi.org/10.1016/B978-0-12-382225-3.00396-0)
[3.00396-0](https://doi.org/10.1016/B978-0-12-382225-3.00396-0), 2015.
